# Carbon dioxide capture and efficient fixation in a dynamic porous coordination polymer

Pengyan Wu [1], Yang Li[1], Jia-Jia Zheng[2,3], Nobuhiko Hosono [2,4], Ken-ichi Otake[2], Jian Wang [1]*, Yanhong Liu[1], Lingling Xia[1], Min Jiang[1], Shigeyoshi Sakaki[3] & Susumu Kitagawa [2]*

Direct structural information of confined $CO_2$ in a micropore is important for elucidating its specific binding or activation mechanism. However, weak gas-binding ability and/or poor sample crystallinity after guest exchange hindered the development of efficient materials for $CO_2$ incorporation, activation and conversion. Here, we present a dynamic porous coordination polymer (PCP) material with local flexibility, in which the propeller-like ligands rotate to permit $CO_2$ trapping. This process can be characterized by X-ray structural analysis. Owing to its high affinity towards $CO_2$ and the confinement effect, the PCP exhibits high catalytic activity, rapid transformation dynamics, even high size selectivity to different substrates. Together with an excellent stability with turnover numbers (TON) of up to 39,000 per $Zn_{1.5}$ cluster of catalyst after 10 cycles for $CO_2$ cycloaddition to form value-added cyclic carbonates, these results demonstrate that such distinctive structure is responsible for visual $CO_2$ capture and size-selective conversion.

[1] School of Chemistry and Materials Science, Jiangsu Key Laboratory of Green Synthetic Chemistry for Functional Materials, Jiangsu Normal University, 221116 Xuzhou, Jiangsu, China. [2] Institute for Integrated Cell-Material Sciences, Kyoto University Institute for Advanced Study, Kyoto University, Yoshida Ushinomiya-cho, Sakyo-ku, Kyoto 606-8501, Japan. [3] Fukui Institute for Fundamental Chemistry, Kyoto University, Nishi-hiraki cho, Takano, Sakyo-ku, Kyoto 606-8103, Japan. [4] Present address: Department of Advanced Materials Science, Graduate School of Frontier Sciences, The University of Tokyo, 5-1-5 Kashiwanoha, Kashiwa, Chiba 277-8561, Japan. *email: wjian@jsnu.edu.cn; kitagawa@icems.kyoto-u.ac.jp

The huge consumption of fossil fuels has resulted in sharply rising levels of anthropogenic $CO_2$ emission, leading to serious linkage problems associated with global warming and climate change[1,2]. Practical $CO_2$ capture and sequestration (CCS) are currently usually limited to extensive energy input during desorption and compression processes[3–5]. Captured $CO_2$ should ideally be synchronously converted into high-value chemicals, allowing the emitted $CO_2$ to be reused in carbon cycling processes. One of the greenest approaches is cycloaddition of $CO_2$ to epoxides, a 100% atom-economical reaction, to form cyclic carbonates, which have numerous applications in petrochemicals, fine chemicals, and pharmaceuticals[6–9]. The relatively low reactivity and kinetic inertness of $CO_2$ make it essential to develop efficient catalytic systems for its incorporation, activation, and conversion. Furthermore, direct structural information of the molecular state of captured $CO_2$ by X-ray diffraction (XRD) techniques is invaluable despite the practical difficulties associated with its gaseous nature because this not only enables a thorough understanding of the specific host–guest interaction or activation mechanism and processes, but also leads to the development of new and improved catalysts. A limited number of reports[10–13] revealed that commensurate $CO_2$-trapping crystal structures were produced when the host has sufficient flexibility to trap guest molecules, resulting in a high probability of developing gas-trapping structures.

Porous coordination polymers (PCPs) or metal-organic frameworks (MOFs) are emerging as a promising class of crystalline porous materials with wide applications, including gas adsorption and storage[14,15] and heterogeneous catalysis[16–18]. PCPs therefore have an advantage over other catalysts with respect to $CO_2$ chemistry because of their integration of the inherent sorptive behavior with the uniform Lewis/Brønsted acidic or basic active sites endowed by their facile tunability and modular nature, as well as their ultra-high surface area and heterogeneous nature[19–21]. Furthermore, flexibility and softness in spatial and electronic structures can provide a dynamic space that transforms in response to guest trapping[22–24]. Flexible structures usually have lower thermal stability than rigid structures in a single network; however, this problem can be overcome by interpenetration[25]. Despite this, the location of $CO_2$ molecules in an interpenetrated PCP crystal has not yet been reported. PCPs also impose size-slective and shape-selective restrictions through readily fine-tuned channels and pores, displaying a molecular sieving effect[26,27]. Thus our aim is to introduce local flexibility, so effectively that $CO_2$ are captured by size and shape-induced fit, and also that all the reactants still possess degree of freedom for the coupling reaction.

Here, we report the successful design of a two-fold interpenetrated framework, Zn-**DPA**·2H$_2$O (**DPA** = 4,4′,4″-tricarboxyltriphenylamine and (E)-1,2-di-(pyridin-4-yl)diazene). Their propeller-like ligands 4,4′,4″-tricarboxyltriphenylamine undergo rotational rearrangement in response to the release and capture of guest molecules, resulting in slight changes of their channels. The PCP shows a high affinity towards $CO_2$ molecules, which is clearly verified by the single-crystal structure of the $CO_2$-adsorbed phases and its high catalytic efficiency and size selectivity with respect to $CO_2$ cycloaddition to epoxides.

## Results

### Synthesis and characterization of Zn-DPA·2H$_2$O and Zn-DPA.
The reaction of Zn(NO$_3$)$_2$·6H$_2$O, 4,4′,4″-tricarboxyltriphenylamine (H$_3$tca) and (E)-1,2-di(pyridin-4-yl)diazene (dpa) under solvothermal condition gave red crystals Zn-**DPA**·2H$_2$O {[Zn$_{1.5}$(tca)(dpa)$_{0.5}$]·2H$_2$O}$_n$ in high yield (82%). A single-crystal XRD study revealed that Zn-**DPA**·2H$_2$O crystallizes in monoclinic space group

$C2/c$. The asymmetric unit of Zn-**DPA**·2H$_2$O contains one and a half Zn$^{2+}$ ions, one deprotonated tca$^{3-}$ anion, half of a dpa ligand and two lattice water molecules. The Zn(1) atom is hex-acoordinated by six carboxylate oxygen atoms belonging to six different tca$^{3-}$ ligands, forming an octahedral geometry. The Zn(2) atom is surrounded by four oxygen atoms from three tca$^{3-}$ ligands and one nitrogen atom from one dpa ligand in a pseudo-tetragonal pyramid geometry (Supplementary Fig. 1). Adjacent Zn(II) centers are linked by carboxylate oxygen atoms of tca$^{3-}$ ligands with a Zn1···Zn2 separation of 3.338(2) Å, forming a trinuclear Zn$_3$(CO$_2$)$_6$ unit (Supplementary Fig. 2). It is noteworthy that two unsaturated zinc centers on both ends of a trinuclear unit are well-oriented toward the channels, facilitating their full accessibility for the substrates to their open sites of Lewis acidic centers (Supplementary Fig. 3).

The tca$^{3-}$ ion resembles a propeller with the central nitrogen having $sp^2$ hybridization, since its phenyl rings are tilted relative to each other with dihedral angles of 78.2(6)°, 80.6°, and 87.6(9)° (Fig. 1a). The dihedral angle between the phenyl rings of two tca$^{3-}$ ligands located at the *trans* position of the Zn$_3$(CO$_2$)$_6$ cluster is 0.0°, and the N···N distance between these ligands is 16.57 and 16.84(3) Å (Supplementary Fig. 5). Each tca$^{3-}$ ligand-linked six Zn(II) centers in a $\mu_6$-$\eta^1$:$\eta^1$:$\eta^1$:$\eta^1$:$\eta^1$:$\eta^2$ manner to form an infinite two-dimensional (2D) sheet parallel to the *ab* plane (Supplementary Fig. 4); these adjacent layers are further pillared in the third dimension by accessorial dpa ligands through their pyridine groups to afford an extended 3D coordination framework with one-dimensional rectangular channels of ca. 13.2 × 9.7 Å$^2$ along the *b*-axis (Supplementary Fig. 6). This exhibits a two-nodal (3,8)-connected **tfz-d** topology with the (4$^3$)$_2$(4$^6$·6$^{18}$·8$^4$) Schläfli symbol (Supplementary Fig. 7). Notably, the large channel allows the penetration of another identical net; thus, the entire structure of Zn-**DPA**·2H$_2$O is a two-fold interpenetrated 3D net (Fig. 1b). The interpenetrated nets are connected through C−H···π interactions (spacing ca. 3.15 Å). Despite interpenetration, three-dimensionally running channels with cross-sections of 5.8 × 11.5 Å$^2$ are still observed along the *b*-axis (Fig. 1c and Supplementary Fig. 3). The solvent-accessible volume calculated by the PLATON program is 1883.3 Å$^3$, which is 31.6% of the unit-cell volume.

The thermogravimetric analysis curve of Zn-**DPA**·2H$_2$O reveals that the lattice water molecules are removed by heating and that the coordination framework is thermally stable up to ca. 260 °C. Thus, desolvation was accomplished by heating as-synthesized Zn-**DPA**·2H$_2$O at 120 °C under vacuum for 32 h to generate desolvated Zn-**DPA** {[Zn$_{1.5}$(tca)(dpa)$_{0.5}$]}$_n$, and TGA (Supplementary Fig. 8) clearly indicated that the guest water molecules were completely excluded. Single-crystal XRD of desolvated Zn-**DPA** indicated that Zn-**DPA** has a similar unit-cell volume with minor distortions ($\Delta V/V_{\text{Zn-DPA·2H}_2\text{O}} = 0.45\%$, $\Delta\beta = +0.4°$) compared to Zn-**DPA**·2H$_2$O (Supplementary Table 1). Three phenyl rings of the tca$^{3-}$ units undergo minor rotational rearrangements upon desolvation, resulting in dihedral angles relative to each other of 80.7(6)°, 81.8(5)°, and 86.6(9)° (Fig. 1d). Two phenyl rings of the tca$^{3-}$ ligands located at the *trans* position around the Zn$_3$(CO$_2$)$_6$ cluster remain parallel to each other (dihedral angle, 0.0°); however, one of N···N distance between these two tca$^{3-}$ ligands decreases to 16.56(5) Å (Supplementary Fig. 5). This also leads to corresponding small changes in many of the key dihedral angles and torsion angles, which are summarized in Supplementary Table 2. In particular, the torsion angle of O−Zn−O−C changes from −3.08° and −56.95° to 1.30° and −58.64°. Detailed analysis revealed small but non-negligible framework deformations. There was a minimal variation in the interlayer distance between the 2D sheets and the angle between the sides of the 3D net, and the distance between the different interpenetrated nets increased

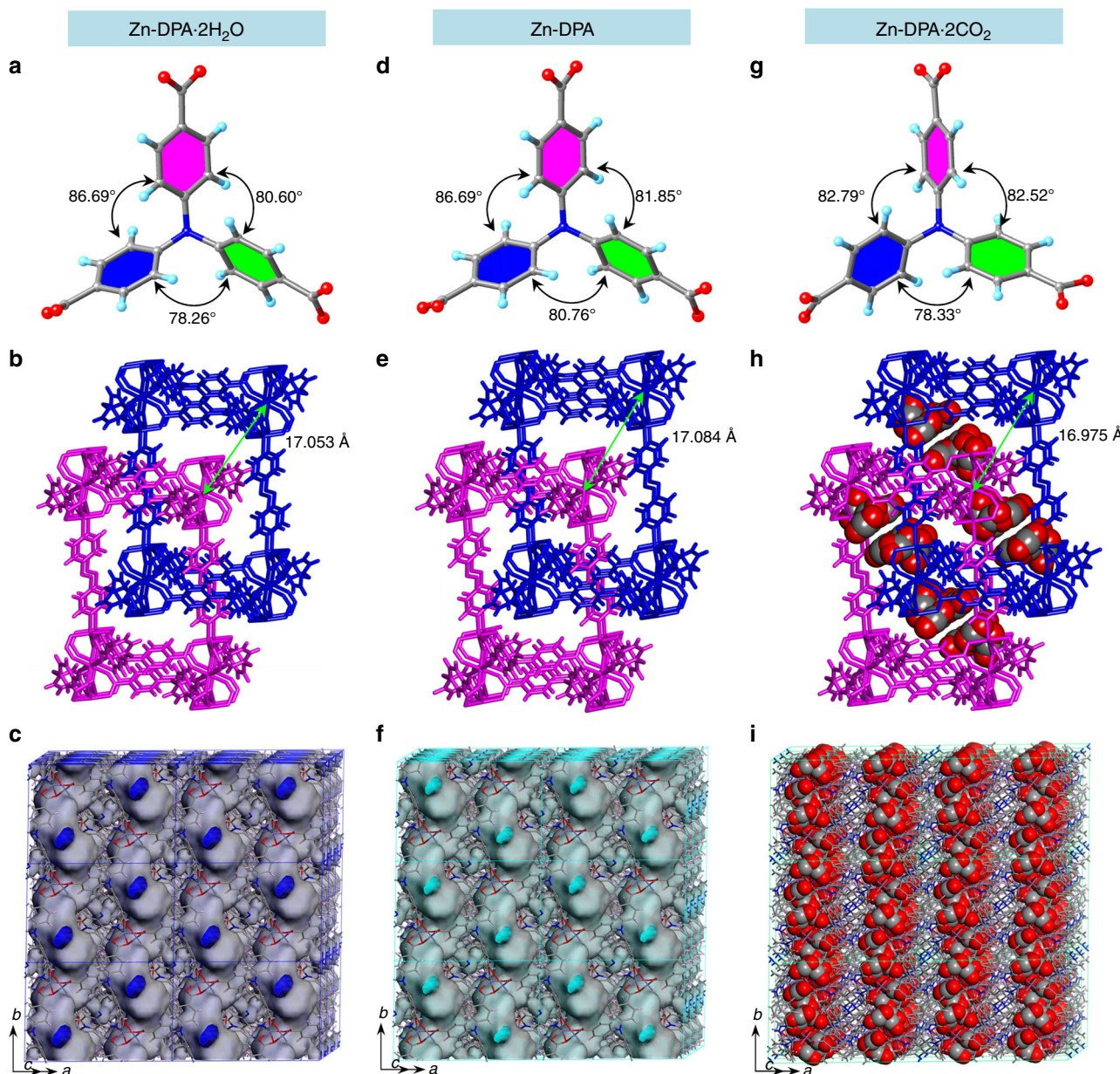

**Fig. 1** X-ray crystal structure analyses. Crystal structures of as-synthesized Zn-**DPA**·2H$_2$O **a–c**, guest-free Zn-**DPA d–f**, and CO$_2$-containing Zn-**DPA**·2CO$_2$ **g–i**. The dihedral angle between two phenyl rings relative to each other in tca$^{3-}$ ligands found in PCPs **a**, **d**, **g**. The two-fold interpenetrated frameworks composed of a trinuclear Zn cluster, tca$^{3-}$, and dpa ligands, showing the interlayer distance between the 2D sheets (measured between adjacent Zn cations from different sheets) **b**, **e**, **h**. Three-dimensionally connected channels in PCPs with Connolly surfaces (Connolly radius: 1.6 Å). The inner surfaces of the channels are shown in blue for **c**, cyan for **f**, and green for **i**, while the outer surfaces are represented in grey. Water molecules are omitted for clarity for Zn-**DPA**·2H$_2$O

slightly from 17.053 Å in Zn-**DPA**·2H$_2$O to 17.084 Å in Zn-**DPA** (Fig. 1e); thus, the main channels (void volume: 31.6%) are still remained. A careful comparison of the two powder X-ray diffraction (PXRD) patterns of as-synthesized Zn-**DPA**·2H$_2$O and Zn-**DPA** also further confirmed that their frameworks remained almost unchanged (Supplementary Fig. 9).

**Direct observation of CO$_2$ molecules trapped in Zn-DPA.** Desolvated Zn-**DPA** shows no obvious sorption of N$_2$ at 77 K (Supplementary Fig. 10). At 195 K, CO$_2$ sorption by Zn-**DPA** gave a reversible Type-I isotherm without hysteresis (Fig. 2a), indicating that the effective micropore filling occurs for CO$_2$. The saturated CO$_2$ uptake was about 78.4 cm$^3$ g$^{-1}$, corresponding to 15.6 wt% or 2.0 CO$_2$ per formula unit. The

Brunauer–Emmett–Teller (BET) surface area of Zn-**DPA** was determined as 283 m$^2$ g$^{-1}$. At 273 and 293 K, the adsorption isotherms of CO$_2$ show a gradual increase and reach maximal amounts of 44.5 cm$^3$ g$^{-1}$ (8.7 wt%) and 34.8 cm$^3$ g$^{-1}$ (6.8 wt%), respectively. The coverage-dependent CO$_2$ adsorption enthalpy ($Q_{st}$) of Zn-**DPA** was calculated using the Clausius–Clapeyron equation based on isotherms measured at 273 and 293 K (Fig. 2b and Supplementary Fig. 11). The obtained $Q_{st}$ was found to be in the range 29.4–32.4 kJ mol$^{-1}$ by fitting the data to the virial model[28]. The high $Q_{st}$ may indicate strong interactions between the Zn-**DPA** coordination framework and CO$_2$ guests.

To determine the interactions between the CO$_2$ adsorbate and the constructed frameworks, a single crystal of evacuated

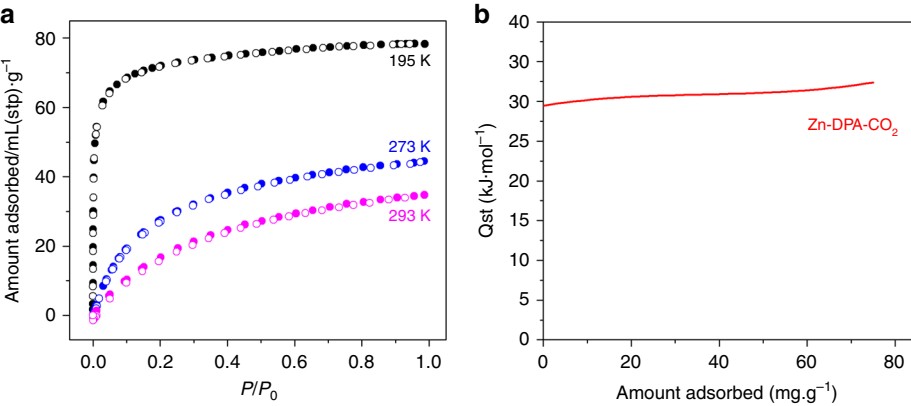

**Fig. 2** Isotherms and isosteric heats of $CO_2$ sorption. **a** $CO_2$ adsorption isotherms for Zn-**DPA** at 195 K (black), 273 K (blue), and 293 K (purple); stp is standard temperature and pressure. **b** Isosteric heats of $CO_2$ adsorption for Zn-**DPA**

Zn-**DPA** was fixed inside a glass capillary and sealed with back-filled $CO_2$ (101 kPa, 195 K). The resulting $CO_2$-loaded single crystal can be characterized with a conventional X-ray diffractometer at 183 K, where Zn-**DPA** could absorb two $CO_2$ molecules per formula unit (Fig. 1h, i). Within each pore, three types of independent $CO_2$-binding sites were located: $CO_2$-I [O(1A)–C(1A)–O(2A)], $CO_2$-II with slight disorder [O(1C)–C(1C)–O(2C), and O(1CC)–C(1CC)–O(2CC)] and $CO_2$-III [O(1B)–C(1B)–O(2B)]. Compared with the structure of the dried Zn-**DPA**, the unit cell of Zn-**DPA**·$CO_2$ displays a slight expansion and an obvious distortion ($\Delta V/V_{Zn-DPA} = 0.34\%$, $\Delta\beta = +0.2°$) (Supplementary Table 1). The dihedral angles of the tca$^{3-}$ ligands are 78.3(3)°, 82.5(2)°, and 82.7(9)° (Fig. 1g). It was found that $CO_2$-I was 77% occupancy, adjacent to the unsaturated zinc center, formed the electrostatic interaction in end-on fashion with a Zn–O distance of 3.17(1) Å, which was significantly shorter than the sum of van der Waals radii of zinc (2.10 Å) and oxygen (1.52 Å). The C($\delta+$) atom of $CO_2$-I also interacts with two carboxylate O atoms from tca$^{3-}$ ligands [C($\delta+$)($CO_2$)···O($\delta-$)(tca) = 3.13(2), 3.22(9) Å]. Another O of $CO_2$ interacts with the pyridyl H atom of dpa ligands and phenyl H atom of tca$^{3-}$ ligands via C–H···O hydrogen bonding with short H···O distances of 2.64(7), 2.88(2) Å, respectively (Supplementary Fig. 12). $CO_2$-II (64% occupancy) and $CO_2$-III (59% occupancy) are located between tca$^{3-}$ ligands via the host–guest C–H···O interactions (2.68(6) Å [O1CC···H4], 2.84(6) Å [O1C···H2], 2.82(1) Å [O2B···H7], respectively). In addition, $CO_2$-I molecule interacts with neighbor $CO_2$-II and $CO_2$-III in T-shaped [(C($\delta+$)···O($\delta-$) = 3.90(9) Å, C1A···O2CC] and slipped parallel conformation [C···C = 3.70(3) Å, C1A···C1B; 3.26(3) Å, C1A···C1C]. The distance between the different interpenetrated nets is 16.975 Å and the void volume is 31.6% (when $CO_2$ molecules are omitted) upon $CO_2$ adsorption (Fig. 1h).

Furthermore, the canonical Monte Carlo (MC) simulations followed by geometry optimization using the PBE-D3 functional[29,30] indicate that $CO_2$ molecules are found at three sites, which agree with the experimental result. The PBE-D3-calculated binding energy (BE) for adsorption of one $CO_2$ molecule in Zn-**DPA** decreases following the order site I > site III ≈ site II, suggesting that the site I is the most favorable for $CO_2$ adsorption at low loading (Supplementary Fig. 13). This result was analyzed by using the $E_{INT}$(H-G) between $CO_2$ and Zn-**DPA** and $E_{DEF}$(H) of Zn-**DPA**. Because $E_{DEF}$(G) is negligibly small, it is not discussed herein. The $E_{DEF}$(H) is similar among the sites I, II, and III (Supplementary Table 3). However, the $E_{INT}$(H-G) at the site I (–9.88 kcal mol$^{-1}$) is much larger (much more negative) than at the sites II and III. This $E_{INT}$(H-G) at the

site I mainly arises from the electrostatic interactions of $CO_2$ molecule with $Zn^{2+}$ and carboxylates because the negatively charged O atom approaches the positively charged Zn atom and the positively charged C atom approaches the negatively charged O atom of the carboxylates (Supplementary Fig. 14). Because the $Zn^{2+}$–$CO_2$ interaction is absent at the sites II and III, the $E_{INT}$(H-G) is weaker at these sites than at the site I. This is the reason why the site I exhibits the largest (most negative) $CO_2$-binding energy. These results indicate that $CO_2$ adsorption into Zn-**DPA** is likely to occur first at the site I and then $CO_2$ adsorption starts to occur at the site II or III. The present calculations also showed the BE value is similar between the sites II and III in the presence of 8 $CO_2$ molecules at the site I (Supplementary Table 4), exhibiting that $CO_2$ adsorption similarly occurs at these sites II and III. These computational results are consistent with the experimental results by the single-crystal XRD that $CO_2$ molecules are found in similar occupancy at both of the sites II and III but the probability is lower than that at the site I. Thus, the combined experimental and computational results suggest that Zn-**DPA** has a high $CO_2$-binding affinity and could activate $CO_2$.

**Size-selective fixation of captured $CO_2$ in Zn-DPA.** The ability of $CO_2$ capture and the Lewis acid metal sites embedded in its framework suggest that Zn-**DPA** is a highly promising size-selective heterogeneous catalyst for reactions of $CO_2$ with substrate. Our catalytic experiments focused on the cycloaddition of carbon dioxide and epoxides. This reaction was performed with Zn-**DPA** in an autoclave reactor at 1 MPa and 373 K. A small aliquot of the supernatant reaction mixture was analyzed by nuclear magnetic resonance spectroscopy ($^1$H NMR) to calculate the reaction yield (Supplementary Fig. 15). As shown in Table 1, the transformation was initially examined by using 2-(phenoxymethyl)oxirane (20 mmol) and $CO_2$ as the coupling partners, along with 5 μmol of Zn-**DPA** (based on $Zn_{1.5}$ cluster) and co-catalytic amounts (0.3 mmol) of tetrabutylammonium bromide (TBABr). The results indicated that Zn-**DPA** can serve as an effective catalyst for the solvent-free synthesis of cyclic carbonates, affording an almost complete conversion within 2 h. The turnover number (TON) was ∼4000 per $Zn_{1.5}$ cluster of Zn-**DPA**, and the turnover frequency (TOF) was ∼2000 per $Zn_{1.5}$ cluster of Zn-**DPA** per hour. To the best of our knowledge, these values are higher than all of those previously reported for MOF-based catalysts in the cycloaddition of carbon dioxide to epoxides under similar conditions[19,31]. Several control experiments were conducted, e.g. the absence of any of the individual components,

**Table 1 Zn-DPA-catalyzed coupling of epoxides with $CO_2$[a]**

| Entry | Epoxides | Molecular Size | Yield[b] (%) | TON[c] | TOF[d] |
|-------|----------|----------------|--------------|--------|--------|
| 1 | | 4.2 Å / 9.4 Å | >99 | 4000 | 2000 |
| 3 | | 3.1 Å / 8.2 Å | 99 | 3960 | 1980 |
| 4 | | 3.1 Å / 9.9 Å | 99 | 3960 | 1980 |
| 5 | | 4.4 Å / 8.6 Å | 98 | 3920 | 1960 |
| 6 | | 3.1 Å / 10.9 Å | 91 | 3640 | 1820 |
| 7 | | 4.2 Å / 10.2 Å | 88 | 3508 | 1754 |
| 8 | | 4.2 Å / 11.6 Å | 81 | 3252 | 1626 |
| 9 | | 5.7 Å / 12.0 Å | 49 | 1940 | 970 |
| 10 | | 6.0 Å / 16.5 Å | 14 | 572 | 286 |

[a]Reaction conditions: epoxide (20 mmol), catalyst (5 μmol, based on $Zn_{1.5}$ cluster), and TBABr (0.3 mmol) under carbon dioxide (1 MPa), 373 K and 2 h
[b]Yield of isolated product was determined from by $^1H$ NMR spectroscopy
[c]Moles of cyclic carbonate per mole of catalyst Zn-**DPA**
[d]Moles of cyclic carbonate per mole of catalyst Zn-**DPA** per hour

containing Zn-**DPA**, TBABr, and both, led to only 10% even hardly any 4-(phenoxymethyl)-1,3-dioxolan-2-one product (Supplementary Table 5, entries 1−3). The subcomponents of Zn-**DPA** were then investigated independently. $H_3$tca, abp, or Zn $(NO_3)_2$·$6H_2O$ instead of Zn-**DPA** was applied to $CO_2$ fixation reaction under the same conditions as mentioned above, leading to 11%, 12%, and 43% yield in this product formation (Supplementary Table 5, entries 4−7). Thus, this demonstrated that our MOF Zn-**DPA** is essential for the cycloaddition reaction of carbon dioxide to epoxides.

We further examined the performance of Zn-**DPA** in corresponding $CO_2$ cycloaddition reactions, under the same mild conditions, with aliphatic epoxides or aromatic epoxides substituted with different functional groups to check the generality of the catalyst. In the presence of aliphatic epoxides, such as allyl glycidyl ether, butyl glycidyl ether, glycidyl methacrylate, and glycol diglycidyl ether, all gave a high yield (> 90%, Table 1, entries 2−5). The introduction of nitro- or methoxy-groups onto the phenyl ring gave ∼88% and 81% of the respective products under the same reaction conditions. Interestingly, as the molecular size of the epoxide substrate increased, the yield of cyclic carbonate decreased significantly, as illustrated by the 49% yield of 4,4′-((1,3-phenylenebis(oxy))bis(methylene))bis(1,3-dioxolan-2-one) (Table 1, Entry 8). With the bulky epoxide bis(4-(oxiran-2-ylmethoxy)phenyl)methane, there was a dramatic decrease in the yield of cyclic carbonates with only

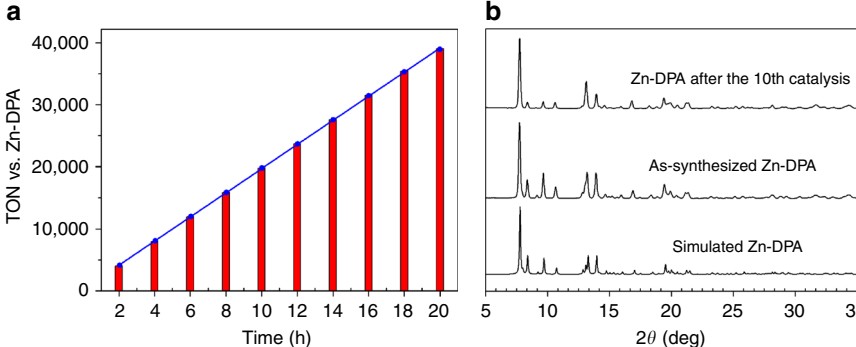

**Fig. 3** Reusability of catalyst Zn-DPA. **a** Time-course and recycling $CO_2$ fixation experiments with Zn-**DPA** under standard conditions. **b** PXRD patterns of simulated Zn-**DPA** (bottom), synthesized Zn-**PDA** (middle), and Zn-**PDA** after 10 successive times of the $CO_2$ fixation reaction with 2-(phenoxymethyl) oxirane (top)

14% conversion of the reactants, suggesting that the large epoxide substrate restricted diffusion into the pores of Zn-**DPA**, which limited the access of reactants to the active sites (Table 1, Entry 9)[11,32], resulting in size-selective catalysis. Such remarkably high efficiency and size selectivity to epoxides in catalytic $CO_2$ cycloaddition confirm that the PCP Zn-**DPA** is a suitable heterogeneous catalyst for carbon fixation. Notably, the catalytic effect in an open-dense flexible PCP was not different for small and bulky epoxide substrates, which highlights the importance of the adaptable channels for the size-selective catalysis for $CO_2$ fixation (Supplementary Table 6).

FT-IR spectra of Zn-**DPA** impregnated with a methanol solution of 2-(phenoxymethyl)oxirane (Supplementary Fig. 16) revealed peaks emerging at 1275 and 920 cm$^{-1}$ corresponding to the characteristic peaks of the Ar–C–O asymmetric and C–O–C symmetric stretching vibrations. A red shift from 1724 cm$^{-1}$ (free epoxide) combined with its $^1$H NMR data (Supplementary Fig. 17) indicated adsorption and activation of the epoxide in the channels of Zn-**DPA**. Density functional theory calculation was carried out to obtain deep insight into the interaction between Zn-**DPA** and 2-(phenoxymethyl)oxirane molecule (Supplementary Fig. 18). The close contact was observed between the epoxy O atom of substrate and the zinc ion (Zn–O: 3.24(4) Å). C–H⋯π interaction was also observed between the phenyl hydrogen atom of the tca$^{3-}$ ligands and phenyl groups of the substrate, with the shortest separation being 3.14(0) Å. The high density of Lewis acid zinc centers confined within the PCP channels can bind the epoxide through its oxygen atom to activate the epoxy ring and also serve as electrostatic binding site to capture $CO_2$[33,34]. Subsequently, the Br$^-$ generated from TBABr attacks the less-hindered methylene C atom of the coordinated epoxide to complete the ring-opening step. This is followed by the interaction of $CO_2$ with the oxygen anion of the opened epoxy ring to form an alkyl carbonate anion, which is then converted to the corresponding cyclic carbonate through the ring-closing step (Supplementary Fig. 20)[35–37]. These combined factors promoted the cycloaddition reaction, resulting in a high catalytic activity of Zn-**DPA** for the chemical conversion of $CO_2$ to cyclic carbonates.

Recyclability is an essential feature of any catalyst considered for use in industrial applications. Therefore, we investigated the catalytic activity of bulk Zn-**DPA** filtered from the catalytic reaction. There was no significant decrease in the efficiency of the catalyst even after 10 cycles of the $CO_2$ fixation reaction with 2-(phenoxymethyl)oxirane, which yielded a total of ~167 mmol product and a total TON of 39,064 per $Zn_{1.5}$ cluster of Zn-**DPA** (Fig. 3a). It is worth noting that Zn-**DPA** gave shorter reaction time and higher product yield than the previously reported

PCPs[19], indicating its suitability for the industrial application in the cycloaddition of the carbon dioxide to cyclic carbonates (Supplementary Table 7). Furthermore, the solid catalyst recovered from the catalytic reaction exhibited the same PXRD pattern as the pristine solid PCP Zn-**DPA** (Fig. 3b), and the IR spectra also support the maintenance of the structure after 10 cycles catalysis (Supplementary Fig. 19), all confirming the stability of the PCP framework during the catalytic reactions. Meanwhile, at the end of the reaction, inductively coupled plasma analysis of the reaction mixture filtrate revealed no Zn leaching, indicating that the catalytic reaction is indeed heterogeneous in nature.

## Discussion

In summary, we have successfully constructed a new two-fold interpenetrated PCP that possesses adaptable property for capturing, concentrating, and converting $CO_2$ using a flexible propeller-like ligand. We have demonstrated that the PCP presents efficient accommodation to $CO_2$, which has been confirmed by single-crystal structure analysis of $CO_2$-adsorbed phases. Its inherent $CO_2$ absorbability, exposed Lewis acid metal sites, and well-defined pores allow this PCP to promote effective size-selective fixation of captured $CO_2$ with appropriate epoxides in a one-pot reaction. Our strategy will open up a new dimension of porous compounds as platforms for determining the gas-loaded crystal structures and efficient utilization of C1 resources. The key to success is to introduce a durable interpenetrated framework, a local flexibility to induce adjustable capture and catalytic Lewis acid sites in the pores.

## Methods

**Characterizations**. Elemental analyses of C, H, and N was performed on a Vario EL III elemental analyzer. Hydrogen NMR spectra were measured on a Bruker-400 spectrometer with Me$_4$Si as an internal standard. X-ray powder diffraction (XRD) patterns of the PCPs were recorded on a Rigaku D/max-2400 X-ray powder diffractometer (Japan) with Cu-Kα ($λ = 1.5405$ Å) radiation. Thermogravimetric analysis (TGA) was carried out at a ramp rate of 5 °C/min in a nitrogen flow with a Mettler-Toledo TGA/SDTA851 instrument. FT-IR spectra were recorded using KBr pellets on a JASCO FT/IR-430 spectrometer. Gas adsorption isotherms were obtained on a BELSORP-max adsorption instrument (BEL Japan Inc.) using a volumetric technique. The initial outgassing of the sample was carried out under high vacuum ($P < 10^{-2}$ Pa, $T = 120$ °C) for 32 h to remove solvated water molecules. The $CO_2$ adsorption isotherms for desolvated compounds were collected in a relative pressure range from 10 to $1.0 \times 10^5$ Pa.

**Synthesis of Zn-DPA·2H$_2$O**. A mixture of 4,4′,4″-tricarboxyltriphenylamine (H$_3$tca) (18.9 mg, 0.05 mmol), (E)-1,2-di(pyridin-4- yl)diazene (9.2 mg, 0.05 mmol), and Zn(NO$_3$)$_2$·6H$_2$O (29.8 mg, 0.1 mmol) was dissolved in ethanol/water (9/1, 5 mL) in a screw capped vial. The resulting mixture was placed in an oven at 110 °C for 3 days. Red block-shaped crystals were obtained after filtration. Yield:

82%. $C_{26}H_{20}N_3O_8Zn_{1.5}$: C 52.00, H 3.36, N 7.00%; Found: C 51.94, H 3.40, N 7.02%.

**Synthesis of Zn-DPA**. Crystals of Zn-**DPA**·2H$_2$O were heated to 120 °C in vacuo for ~32 h to afford single crystals of desolvated Zn-**DPA** suitable for single-crystal XRD. Anal. calcd. for $C_{26}H_{16}N_3O_6Zn_{1.5}$: C, 55.31; H, 2.86; N, 7.44%. Found: C, 55.36; H, 2.79; N, 7.43%.

**Single-crystal X-ray diffraction analyses**. A single crystal was selected to put in a capillary and evacuated at 120 °C under reduced pressure (below 10$^{-2}$ Pa) overnight. Then, CO$_2$ was slowly introduced into the capillary until the pressure reaching 101 kPa at 195 K. After 10 min at 195 K under 101 kPa CO$_2$ atmosphere, the glass capillary was sealed using small torch flame. The gas-loaded crystals were mounted onto a Rigaku XtaLAB AFC10 diffractometer equipped with Mo K$\alpha$ ($\lambda = 0.71073$ Å) optic and slowly cooled down and diffraction data were collected at 183 K. Solution and refinement of this structure were performed through direct methods and full-matrix least-squares methods based on $F^2$ values with SHELXTL version 2018/3.

**Typical procedure for CO$_2$ cycloaddition of epoxides**. The catalytic reaction was conducted in a 30 mL autoclave reactor, which was purged with 1 MPa CO$_2$ under constant pressure for 15 min to allow the system equilibration. The vessel was set in an oil bath with frequent stirring at 373 K for 2 h. At the end of the reaction, the reactor was placed in an ice bath for 20 min and then opened. The catalysts were separated by centrifugation, and a small aliquot of the supernatant reaction mixture was analyzed by $^1$H NMR to calculate the reaction yields.

## Data availability

The X-ray crystallographic coordinates for structures reported in this study have been deposited at the Cambridge Crystallographic Data Centre (CCDC), under deposition numbers 1564761–1564763. These data can be obtained free of charge from The Cambridge Crystallographic Data Centre via www.ccdc.cam.ac.uk/data_request/cif. All other data supporting the findings of this study are available within the article and its Supplementary Information, or from the corresponding author upon reasonable request.

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

## Acknowledgements

This work was supported by the Natural Science Foundation of Jiangsu Province for Outstanding Youth (grant number BK20180105), the National Natural Science Foundation of China (grant numbers 21401087, 21401086), PAPD and TAPP of Jiangsu Higher Education Institutions, JSPS KAKENHI Grant-in-Aid for Specially Promoted Research (Grant no. 25000007) and Scientific Research (S) (JP18H05262). We also thank the support from WPI-iCeMS.

## Author contributions

J.W. and P.W. conceived and directed the project. P.W. and J.W. prepared and analyzed all compounds and carried out the gas sorption. Y.Li and Y.Liu performed catalytic measurements. J.-J.Z. and S.S. performed theoretical calculation. N.H. and K.O. assisted the crystallographic study. L.X. and M.J. assisted the PCP synthesis. P.W., J.W. and S.K. co-wrote the paper. All the authors discussed the results and commented on the manuscript.

## Competing interests

The authors declare no competing interests.
