## [Transparent Peer Review File · Nature Communications]

Reviewers' comments:

Reviewer #1 (Remarks to the Author):

First of all, I think this paper is a tale of 2 halves. The underlying principals behind the manufacture of a interpenetrated framework for CO₂ trapping, and flexibility is an interesting angle, and certainly the cycloaddition results presented are really very nice, however I have several main reservations about this manuscript. The main issue is that the authors give no detail about how the single-crystal data collection was conducted for the CO₂ containing framework. Structural changes given are very small, and whether these are statistically significant or not, has not been mentioned or discussed at all. The corresponding changes in torsion (or dihedral) angles is unclear, as is their significance on the affect these changes have on the structure, both on desolvation, and on uptake of CO₂ (which again, as stated above is not described).

Much more detail is needed here, was CO₂ uptake 100%, was an in-situ X-ray diffraction setup used? What was the pressure of CO₂ applied? Were restraints applied to the data? .fcf files were also not given with, or are included in the .cif files attached as supplementary information, either were restraints and constraints used during refinement included in the .cifs. This is completely unacceptable. If this was submitted to an IUCr journal, it would have been sent back to the authors before being sent to reviewers.

In addition, the affinity, or strong binding of CO₂ in this framework needs to be quantified via computational or another technique. This is pivotal to the reactivity of the framework to be used in the cycloaddition reactions presented here, but appears to be glossed over.

Overall, nice chemistry but much more needed to finish the story, and clarify to the reader what makes this particular MOF good for this type of catalytic chemistry. I would stringly recommend rejecting this manuscript, and suggest the authors address the points I have raised, before submitting to any journal.

Below are some general comments, some of which have already been given above;

In Figure 1., the dihedral angle looks more like an angle drawn. Could this diagram and the actual angle of interest be better displayed?

Space between 'the phenyl' on page 7

Changes in dihedral angles around the tca³⁻ ligands and coordination bond angles are mentioned, but are they statistically significant?

How was the CO₂ structure collected? In-situ gas cell at a synchrotron? More information about the gas pressure etc. is needed here in the main manuscript. Its not until page 10 that it becomes apparent that this is in-fact PXRD data, but then in the conclusions state that this is confirmed by SXD. This needs to be clarified. In-fact, the PXRD patterns appear to be those simulated by the SXD data, but are talked about as if they were experimental data in the manuscript, but these could be the data collected on the Rigaku powder diffractometer. This is very misleading, confusing and poorly written.

Expansion is again mentioned on uptake of CO₂, are the changes in volume statistically significant?

Page 9, 'b axis' should read 'b-axis'

Where is the evidence that the CH...O interaction is a stabilising interaction? A recent paper by Parsons et al., on glycine showed that sometimes even quite obvious looking NH...O interactions can in-fact be de-stabilising (CrystEngComm, 2015, 17, 5315-5328). I'd expect to see some kind of computational back up to such a broad statement.

More detail is also needed on the search criteria used to search for the 3-3.4 Å range quoted for the range of H...O distances. The paper references is form 2014. Since then, a dedicated MOF-database has been included in the CSD, with many more structures added. I would expect the authors to conduct their own search for close interactions to make sure that the broad statement they are making here is in-fact true. In addition, no e.s.d.'s are included for the H...O distance (quoted as 2.597 Å), which is a necessary requirement.

Love the work on cycloadditon reactions, and the size selectivity shown by the authors, this is really nice work, but the descriptions, and crystallographic data presented at the start of the manuscript severely let this down.

Reviewer #2 (Remarks to the Author):

In this manuscript, the authors showed a porous framework with high affinity towards CO₂ molecules. The interaction between host and CO₂ has also been confirmed by single-crystal structure analysis of CO₂-adsorbed phases. The obtained results raise some interesting points on CO₂ trapping. However, the discussion on the cycloaddition of carbon dioxide and epoxides is quite insufficient, and the catalytic mechanism is immature. The following points should be addressed, and some discussion needs to be rewritten.

1. N₂ sorption measurement and pore size distribution curve for Zn-DPA have not been shown in the manuscript. It seems that the molecular sizes of all epoxides substrate and their corresponding product are much larger than the narrow pores in Zn-DPA.
2. The saturated CO₂ uptake corresponds to two CO₂ per formula unit, however, X-ray single crystal diffractometer data indicates Zn-DPA could absorb one CO₂ molecule per formula unit. Please explain why these results are inconsistent.
3. The authors claim that the porosity and synergistic interaction of Zn-DPA favors CO₂ uptake and further increases the local concentration of CO₂ in the channel, increasing the collision frequency of activated CO₂ with activated epoxide intermediates. However, there is no experimental evidence to support their verdict, and the epoxide can't diffuse into the narrow pores in Zn-DPA. Please give a more reasonable explanation.
4. Lewis acid sites inside MOFs are necessary for the cycloaddition of carbon dioxide and epoxides. How, in the structure, Zn(1) and Zn(2) atoms respectively show octahedral and trigonal bipyramidal geometries and no open metal site as Lewis acid is reserved for catalysis.
5. In my opinion, size selectivity for epoxides substrate is not an advantage for Zn-DPA, and goes against the extension of substrate scope in this chemical CO₂ fixation reaction. Please extend the substrate scope of this chemical CO₂ fixation reaction catalyzed by Zn-DPA.
6. No controlling experiment has been carried out to attest the significance of MOF catalyst during catalytic process.
7. What about the reusability of Zn-DPA catalyst? The PXRD pattern of Zn-DPA sample after the catalyst is missed in the manuscript.

Reviewer #3 (Remarks to the Author):

This report synthesized a crystalline porous structure called Zn-DPA, using two ligands of H₃tca and dpa, which was characterized to be interpenetrated and exhibited a high affinity to CO₂, as determined by adsorption isotherms and single-crystal XRD, and demonstrated the catalysis results with different sized substrates.

The following questions should be considered in revision of the paper.

1. The authors claimed that the flexibility and rigidity of the interpenetrated frameworks are responsible for the CO₂ trapping and size-selectivity in catalysis, respectively. Can the authors give direct evidence to support this point? Or what if using H₃tca and dpa as ligand separately to synthesize PCPs ?
2. The molecular size of the substrates used should be compared with the channels of the PCP catalysts.
3. Since many other interpenetrated PCPs already exist in the literature, what is the unique features of this proposed Zn-DPA endowing their properties in fixing CO₂? The authors should provide more insightful discussions on the design rationales.

Point-by-point response to the referees' comments

The reviewers' comments are written with *Italic font* and replies are written in blue.

Reply to the Reviewer #1's comments

First of all, I think this paper is a tale of 2 halves. The underlying principles behind the manufacture of an interpenetrated framework for CO₂ trapping, and flexibility is an interesting angle, and certainly the cycloaddition results presented are really very nice, however I have several main reservations about this manuscript. The main issue is that the authors give no detail about how the single-crystal data collection was conducted for the CO₂ containing framework. Structural changes given are very small, and whether these are statistically significant or not, has not been mentioned or discussed at all. The corresponding changes in torsion (or dihedral) angles is unclear, as is their significance on the affect these changes have on the structure, both on desolvation, and on uptake of CO₂ (which again, as stated above is not described).

Response: We have added a section of "Single-crystal X-ray diffraction analyses" in "Methods" in page 16 to detail the setup for the CO₂-loaded single-crystal data collection.

We carefully revised the manuscript to discuss the details of the structural transformation based on desolvation and uptake of CO₂, including the corresponding changes in torsion (or dihedral) angles summarized in Supplementary Table S3. In fact, we added the relative information as follows.

Page 6, line 24: *"This also leads to corresponding small changes in many of the key dihedral angles and torsion angles, which are summarized in Table S3. In particular, the torsion angle of O–Zn–O–C changes from 10.92° and 57.22° to 0.33° and –59.61°. Detailed analysis revealed small but non-negligible framework deformations."*

Page 8, line 12: *"The dihedral angles of the tca³⁻ ligands are 79.1(7)°, 79.9(0)°*

and 85.8(5)° (Figure 1g). Many relative dihedral angles and torsion angles changed accordingly (Supplementary Table S3).”

Much more detail is needed here, was CO₂ uptake 100%, was an in-situ X-ray diffraction setup used? What was the pressure of CO₂ applied? Were restraints applied to the data? .fcf files were also not given with, or are included in the .cif files attached as supplementary information, either were restraints and constraints used during refinement included in the .cifs. This is completely unacceptable. If this was submitted to an IUCr journal, it would have been sent back to the authors before being sent to reviewers.

Response: We appreciate the reviewer’s comments on the experimental setup for the CO₂-trapping crystal structure. On response to the reviewer, we carried out the single crystal X-ray diffraction experiments and characterized the structures well. We have added “Single-crystal X-ray diffraction analyses” section in page 16 to show the details for CO₂-trapping crystal preparation, examination and refinement. Accordingly, we also revised our manuscript to make the examination condition clearer as follows.

Page 8, line 2: *“To determine the interactions between the CO₂ adsorbate and the constructed frameworks, a single crystal of evacuated Zn-DPA was fixed inside glass capillaries and sealed with back-filled CO₂ (95 kPa, 293 K).”*

We have re-resolved the crystal data using the new version of ShelXL, which automatically embeds the res and the hkl file into the CIF and makes it easier to provide the related refinement information. During refinement, CO₂-I [O(4)-C(2)-O(8)] could be anisotropically refined without any restriction. For CO₂-II [O(103)-C(102)-O(104)] and CO₂-III [O(105)-C(103)-O(106)], the C=O bond distances were fixed at 1.15 Å. The new CIF files and checkCIF reports are uploaded.

In addition, the affinity, or strong binding of CO₂ in this framework needs to be quantified via computational or another technique. This is pivotal to the reactivity of the framework to be used in the cycloaddition reactions presented here, but appears to

be glossed over.

Response: We appreciate the reviewer's suggestion. In our revised manuscript, we performed theoretical calculations to understand the strong binding of CO₂ with this framework. In addition to the experimentally determined CO₂ adsorption position (site I), we found other two possible CO₂ adsorption sites (site II and III) in Zn-DPA through Monte-Carlo simulation followed by using DFT calculation geometry optimization. The calculated binding energy for adsorption of one CO₂ molecule decreases in the order of site I > site II ≈ site III, suggesting that CO₂ adsorption prefers occurring at the site I first and has similar probability to occur at the sites II and III. We made further analysis using several cluster models to understand which part of Zn-DPA contributes mostly to the strong CO₂ binding energy at these sites. Such analysis shows that the electrostatic interaction between O of CO₂ and Zn²⁺ (open-metal site) and the electrostatic interaction between C(δ⁺) of CO₂ and O(δ⁻) of tca³⁻ carboxylate play crucially important role for the strong CO₂ binding energy at the site I. Detailed explanations on these computational results are presented in the manuscript (see page 9-10) and supporting information (Section Computational Details, Tables S4 and S5, Figures S13 and S14, and the corresponding discussion).

Overall, nice chemistry but much more needed to finish the story, and clarify to the reader what makes this particular MOF good for this type of catalytic chemistry. I would strongly recommend rejecting this manuscript, and suggest the authors address the points I have raised, before submitting to any journal.

Response: We have revised our manuscript according to the comments. We hope our revised manuscript is now suitable for publication in *Nature Communications*.

Below are some general comments, some of which have already been given above; In Figure 1., the dihedral angle looks more like an angle drawn. Could this diagram and the actual angle of interest be better displayed?

Response: Sorry not to mark these dihedral angles clear to cause confusion in

Figure 1, we added the plane of the phenyl ring of tca^{3-} ligands to better display dihedral angles around the tca^{3-} ligands.

Space between 'the phenyl' on page 7

Response: The typo was corrected as “the phenyl” on page 5.

Changes in dihedral angles around the tca^{3-} ligands and coordination bond angles are mentioned, but are they statistically significant?

Response: There are small changes in dihedral angles around the tca^{3-} ligands and many of the key dihedral angles and torsion angles upon desolvation and CO_2 uptake, which are summarized in Table S3. Detailed analysis revealed small but non-negligible framework deformations.

How was the CO_2 structure collected? In-situ gas cell at a synchrotron? More information about the gas pressure etc. is needed here in the main manuscript. It's not until page 10 that it becomes apparent that this is in-fact PXRD data, but then in the conclusions state that this is confirmed by SXD. This needs to be clarified. In-fact, the PXRD patterns appear to be those simulated by the SXD data, but are talked about as if they were experimental data in the manuscript, but these could be the data collected on the Rigaku powder diffractometer. This is very misleading, confusing and poorly written.

Response: We really appreciate the reviewer's comments and added the detailed information about CO_2 -loaded single crystal collection in the "Methods" of the revised manuscript. First, a good single crystal of dried Zn-DPA was selected to put in a capillary and evacuated under reduced pressure (below 10^{-2} Pa) for one hour at 120 °C. Then the crystal was backfilled with CO_2 (95 kPa, 293 K) on volumetric adsorption apparatus (Bel-max) and allowed to equilibrate for 10 mins before being flame sealed. Secondly, the gas loaded crystals were mounted onto a Rigaku Saturn 70 CCD diffractometer equipped with Mo $\text{K}\alpha$ ($\lambda = 0.71073$ Å) optic and confocal

monochromator and slowly cooled down and diffraction data were collected at 123 K. Solution and refinement of this structure were performed through direct methods and full-matrix least-squares methods based on F^2 values with SHELXTL version 2018/3.

Sorry to make confusion on understanding CO₂-included structure using PXRD patterns, thus we deleted this description. Instead, CO₂-included structure analysis is used directly to indicate the framework changes upon CO₂ uptake. The resulting CO₂-loaded single crystal analysis revealed that Zn-DPA could absorb 1.27 CO₂ molecule per formula unit (Figure 1h and Figure 1i). Within each pore, three types of independent CO₂ binding sites were located: CO₂-I [O(4)-C(2)-O(8)], CO₂-II [O(103)-C(102)-O(104)] and CO₂-III [O(105)-C(103)-O(106)] (Figure 2b inset). Compared with the structure of the dried Zn-DPA, the unit cell of Zn-DPA·1.27CO₂ displays a slight expansion and an obvious distortion ($\Delta V/V_{\text{Zn-DPA}} = 0.3\%$, $\Delta\beta = +1.1^\circ$) (Supplementary Table S2). It is noted that the distance (17.070 Å) between the different interpenetrated nets is almost unchanged (17.058 Å for Zn-DPA) and the main channels (void volume: 30.9% when CO₂ molecules are omitted) are still maintained upon CO₂ adsorption (30.5% for Zn-DPA) (Figure 1h), further highlighting the rigid character of Zn-DPA channels at the same time.

Expansion is again mentioned on uptake of CO₂, are the changes in volume statistically significant?

Response: Compared with the structure of the dried Zn-DPA, the unit cell of Zn-DPA·1.27CO₂ displays a slight expansion and an obvious distortion ($\Delta V/V_{\text{Zn-DPA}} = 0.3\%$, $\Delta\beta = +1.1^\circ$). Within each pore, three types of independent CO₂ binding sites were located. This framework deformation is small but non-negligible, revealing that the flexible character of Zn-DPA is just poised for transforming to permit CO₂ trapping.

Page 9, 'b axis' should read 'b-axis'

Response: The “b axis” was corrected as “b-axis” on page 6.

Where is the evidence that the CH...O interaction is a stabilising interaction? A recent paper by Parsons et al., on glycine showed that sometimes even quite obvious looking NH...O interactions can in-fact be de-stabilising (CrystEngComm, 2015,17, 5315-5328). I'd expect to see some kind of computational back up to such a broad statement.

More detail is also needed on the search criteria used to search for the 3-3.4 Å range quoted for the range of H...O distances. The paper reference is from 2014. Since then, a dedicated MOF-database has been included in the CSD, with many more structures added. I would expect the authors to conduct their own search for close interactions to make sure that the broad statement they are making here is in-fact true. In addition, no e.s.d.'s are included for the H...O distance (quoted as 2.597 Å), which is a necessary requirement.

Response: We appreciate the reviewer for pointing out this problem. From the CO₂-loaded crystal structure analysis, we can find that there are three kinds of interactions around CO₂-I. (1) one O of CO₂ was adjacent to the unsaturated zinc center, forming the electrostatic interaction in end-on fashion with a Zn–O distance of 3.18(2) Å. (2) The C(δ+) atom of CO₂-I also interacts with two carboxylate O atoms from tca³⁻ ligands [C(δ+)(CO₂)...O(δ-)(tca) = 3.16(7) Å, 3.31(2) Å]. (3) Another O of CO₂ interacts with the pyridyl H atom of dpa ligands and phenyl H atom of tca³⁻ ligands *via* C–H...O hydrogen bonding with short H...O distances of 2.59(2) Å, 2.86(7) Å, respectively (Supplementary Fig. S12). In addition, we carried out computational studies which suggest that the C–H...O(CO₂) interaction is indeed attractive. For instance, the interaction energy of CO₂ with SCM^{A3} is –1.06 kcal/mol (Supplementary Fig. S14), where the C–H...O interaction is presented. However, this type of interaction is not the main reason for the strong CO₂ binding ability of this PCP. We therefore revised the corresponding discussion in our manuscript (see page 8, lines 20-23 in the main text).

Love the work on cycloaddition reactions, and the size selectivity shown by the authors, this is really nice work, but the descriptions, and crystallographic data presented at the start of the manuscript severely let this down.

Response: We really appreciate such encouraging remarks on our work about cycloaddition reactions by the reviewer. According to the reviewer's suggestions, we carefully revised the manuscript to discuss the details of the structural transformation based on desolvation and uptake of CO₂ and illuminate the strong binding of CO₂ in this framework through theoretical calculations. Such direct observation of carbon dioxide trapping in a porous catalyst is of great help to explain the mechanism of the following CO₂ cycloaddition reaction. On the one hand, we can clearly find that the high density of Lewis acid zinc centres confined within the PCP channels can bind the epoxide through its oxygen atom to activate the epoxy ring and also serve as electrostatic binding sites to capture CO₂. On the other hand, the maintenance of its channels even upon desolvation can impose size- and shape-selective restrictions for substrates during the catalytic process. These combined factors promoted the cycloaddition reaction, resulting in a high catalytic activity of Zn-DPA for the chemical conversion of CO₂ to cyclic carbonates.

Reply to the Reviewer #2's comments

In this manuscript, the authors showed a porous framework with high affinity towards CO₂ molecules. The interaction between host and CO₂ has also been confirmed by single-crystal structure analysis of CO₂-adsorbed phases. The obtained results raise some interesting points on CO₂ trapping. However, the discussion on the cycloaddition of carbon dioxide and epoxides is quite insufficient, and the catalytic mechanism is immature. The following points should be addressed, and some discussion needs to be rewritten.

1. N₂ sorption measurement and pore size distribution curve for Zn-DPA have not been shown in the manuscript. It seems that the molecular sizes of all epoxides substrate and their corresponding product are much larger than the narrow pores in

Zn-DPA.

Response: Thanks for the reviewer's kind suggestion. We measured N₂ adsorption isotherm for *Zn-DPA* at 77 K (Supplementary Fig. S10). As shown in Figure S10, the N₂ adsorption amount is negligibly small, probably because of the small binding energy of N₂ in this PCP. It is calculated according to the same method as CO₂ binding energy, in which the computational details are listed in the revised supporting information and the calculation result is shown in Table S9 below. Accordingly we added the information “*Desolvated Zn-DPA shows no obvious sorption of N₂ at 77 K (Supplementary Fig. S10).*” in the revised manuscript. On the other hand, computational study for epoxide substrate (2-(phenoxyethyl)oxirane) also show that its binding energy is much more negative than that of CO₂. As a result, the accommodation of epoxide substrate is possible from the thermodynamic viewpoint. We also compared the pore size of this PCP and molecular sizes of the epoxides substrate (Table 1 in the revised manuscript), which shows that the first eight epoxides substrate are (slightly) smaller than the pore size, suggesting that these epoxides can diffuse into and out of the pore of this PCP.

Table S9. Binding Energy (BE, kcal mol⁻¹) for adsorption of one CO₂, N₂, and epoxide in *Zn-DPA*.

Species	CO ₂	N ₂	Epoxide
BE	-9.28	-5.99	-33.92

Supplementary Figure S10. Gas sorption isotherms of Zn-DPA for CO₂ (195 K) and N₂ (77 K).

2. The saturated CO₂ uptake corresponds to two CO₂ per formula unit, however, X-ray single crystal diffractometer data indicates Zn-DPA could absorb one CO₂ molecule per formula unit. Please explain why these results are inconsistent.

Response: We appreciate the reviewer's comments. We have resolved carefully the CO₂-included crystal data and found that Zn-DPA could absorb 1.27 CO₂ molecules per formula unit (Figure 1h and Figure 1i). Within each pore, three types of independent CO₂ binding sites were located: CO₂-I [O(4)-C(2)-O(8)] with 100% occupancy, CO₂-II [O(103)-C(102)-O(104)] with 11% occupancy and CO₂-III [O(105)-C(103)-O(106)] with 16% occupancy (Figure 2b inset).

Furthermore, we performed theoretical calculations to understand the CO₂ binding at those three sites in this framework. As shown in Table S4, the PBE-D3-calculated binding energy (BE) for adsorption of one CO₂ molecule in Zn-DPA decreases in the order site I > site III ≈ site II, suggesting that site I is the most favorable for CO₂ adsorption at low loading. To understand the reason, the BE was decomposed into the interaction energy (E_{INT}) between CO₂ and Zn-DPA framework and deformation energy (E_{DEF}) of Zn-DPA framework. Because the deformation energy of CO₂ is

negligibly small, it will not be discussed herein. The deformation energy of Zn-DPA is similar among CO₂ adsorptions at the sites I, II, and III (Supplementary Table S4). However, the interaction energy of CO₂ with Zn-DPA at the site I is $-9.88 \text{ kcal mol}^{-1}$, much larger (more negative) than at the sites II and III. The CO₂ interaction energy at the site I mainly arises from the electrostatic interactions of CO₂ molecule with Zn²⁺ and carboxylates, which agrees well with CO₂-loaded crystal structure (Fig. S14). Because such CO₂-Zn²⁺ interaction is absent at the sites II and III, the $E_{\text{INT}}(\text{H-G})$ is weaker at these sites than at the site I. As a result, the site I has the largest (most negative) CO₂ binding energy, indicating that CO₂ adsorption into Zn-DPA is likely to occur at the site I first and then CO₂ adsorption starts to occur at the site II or III after the site I is fully occupied. Further calculation showed the BE values are similar between CO₂ adsorptions at the sites II and III (Table S5), suggesting that these two sites have similar probabilities to be occupied by CO₂ molecules, which is consistent with the experimental results that CO₂ molecule with similar occupancy can be found at both sites II and III according to the single-crystal XRD data.

3. The authors claim that the porosity and synergistic interaction of Zn-DPA favors CO₂ uptake and further increases the local concentration of CO₂ in the channel, increasing the collision frequency of activated CO₂ with activated epoxide intermediates. However, there is no experimental evidence to support their verdict, and the epoxide can't diffuse into the narrow pores in Zn-DPA. Please give a more reasonable explanation.

Response: According to the reviewer's comments, we revised the section "Reason analysis for high catalytic activity of Zn-DPA for the chemical conversion of CO₂ to cyclic carbonates" in the main text. We re-analyzed the CO₂-loaded single crystal structure and performed theoretical calculation for the binding energy of CO₂ and epoxide substrates in this framework. All the data showed that Lewis acid zinc centres confined within the PCP channels can bind the epoxide through its oxygen atom to activate the epoxy ring and also serve as electrostatic binding sites to capture CO₂.

Accordingly we revised these relative discussions in this section. (see page 12)

4. Lewis acid sites inside MOFs are necessary for the cycloaddition of carbon dioxide and epoxides. How, in the structure, Zn(1) and Zn(2) atoms respectively show octahedral and trigonal bipyramidal geometries and no open metal side as Lewis acid is reserved for catalysis.

Response: As pointed out by the reviewer, in our PCP structure, the Zn(2) atom is hexacoordinated by six carboxylate oxygen atoms belonging to six different tca^{3-} ligands, forming an octahedral geometry; the Zn(1) atom is five-coordinated by four oxygen atoms from three tca^{3-} ligands and one nitrogen atom from one dpa ligand in a distorted trigonal bipyramidal geometry. The distortion of this trigonal bipyramidal geometry tends to pseudo-tetragonal pyramid configuration. And adjacent Zn(II) centers are linked by carboxylate oxygen atoms of tca^{3-} ligands to form a trinuclear $\text{Zn}_3(\text{CO}_2)_6$ unit. In this case, zinc centers on both ends of a trinuclear unit are well-oriented toward the channels, thus having open space to access the substrate molecules as Lewis acidic sites. Accordingly we have revised Zn(1) geometry into “pseudo-tetragonal pyramid geometry” in the structure section and added the following sentence in page 5, line 13: “It is noteworthy that two unsaturated zinc centers on both ends of a trinuclear unit are well-oriented toward the channels, facilitating their full accessibility for the substrates to their open sites of Lewis acidic centers.”

Supplementary Figure S1. Coordinated environment of the Zn^{2+} ions in $\text{Zn-DPA}\cdot 2\text{H}_2\text{O}$. The right picture represents the enlarged view of pseudo-tetragonal pyramid configuration of Zn(1) center. Symmetry codes: A -x, -y, -z.

Supplementary Figure S3. Three-dimensionally connected channels in Zn-DPA·2H₂O with the Connolly surfaces (Connolly radius: 1.6 Å) along *b*-axis. The yellow arrows indicate the free Lewis acid zinc centers.

5. *In my opinion, size selectivity for epoxides substrate is not an advantage for Zn-DPA, and goes against the extension of substrate scope in this chemical CO₂ fixation reaction. Please extend the substrate scope of this chemical CO₂ fixation reaction catalyzed by Zn-DPA.*

Response: According to the reviewer's comments, we have further extended the substrate scope of this chemical CO₂ fixation reaction catalyzed by Zn-DPA. Besides the aromatic epoxides, *aliphatic epoxides* (such as allyl glycidyl ether, butylglycidyl ether, glycidyl methacrylate and glycol diglycidyl ether) were also catalyzed as the substrates in the same condition, resulting in a high yield (> 90%, Table 1, Entries 2–5). These results were added in Table 1 of the revised manuscript. All the data demonstrated that our catalyst Zn-DPA had a good generality for such CO₂ cycloaddition.

6. *No controlling experiment has been carried out to attest the significance of MOF catalyst during catalytic process.*

Response: Thanks for the kind suggestion. Several control experiments were conducted, *e.g.* the absence of any of the individual components, containing Zn-DPA,

TBABr and both, led to only 10% even hardly any 4-(phenoxymethyl)-1,3-dioxolan-2-one product (Table S6, entries 1–3). The subcomponents of Zn-DPA were then investigated independently. H₃tca, abp or Zn(NO₃)₂·6H₂O instead of Zn-DPA was applied to CO₂ fixation reaction under the same conditions as mentioned above, leading to 11%, 12% and 43% yield in this product formation. (Table S6, entries 4–7) Thus this demonstrated that our MOF Zn-DPA is essential for the cycloaddition reaction of carbon dioxide to epoxides. These related data were added in the revised supporting information, and the related discussion was also added in the revised manuscript (see page 10).

Supplementary Table S6. Control experiments of coupling of 2-(phenoxymethyl)oxirane with CO₂.^a

Entry	Catalyst	Co-catalyst	Yield(%) ^c	TON ^d	TOF ^e
1	none	none	n.d.	n.d.	n.d.
2	Zn-DPA	none	10	400	200
3	none	TBABr	12		
4 ^b	H ₃ tca	TBABr	12	480	240
5 ^b	abp	TBABr	12	960	480
6 ^b	Zn(NO ₃) ₂ ·6H ₂ O	none	11	294	147
7 ^b	Zn(NO ₃) ₂ ·6H ₂ O	TBABr	43	1146	573
8	Zn-DPA	TBABr	> 99	4000	2000

^a Reaction conditions: 2-(phenoxymethyl)oxirane (20 mmol), catalyst (5 μmol, based on Zn_{1.5} cluster) and TBABr (0.3 mmol) under carbon dioxide (1 MPa), 373 K and 2 h. ^b The amount of subcompounds: H₃tca (5 μmol), abp (2.5 μmol), Zn(NO₃)₂·6H₂O (7.5 μmol), confirming the same concentration as these units of Zn-DPA. ^c Yield of isolated product was determined from by ¹H NMR spectroscopy. ^d Moles of cyclic carbonate per mole of catalyst. ^e Moles of cyclic carbonate per mole of catalyst per hour.

7. *What about the reusability of Zn-DPA catalyst? The PXRD pattern of Zn-DPA sample after the catalyst is missed in the manuscript.*

Response: Recyclability is an essential feature of any catalyst considered for use in industrial applications. Therefore, we investigated the catalytic activity of bulk Zn-DPA filtered from the catalytic reaction. There was no significant decrease in the efficiency of the catalyst even after 10 cycles of the CO₂ fixation reaction with 2-(phenoxy)methyl)oxirane, which yielded a total of approximately 167 mmol product and a total TON of 39,064 per Zn_{1.5} cluster of Zn-DPA (Figure 3a). Furthermore, the solid catalyst recovered from the catalytic reaction exhibited the same PXRD pattern as the pristine solid PCP Zn-DPA (Figure 3b), meanwhile, the IR spectra also support the maintain of the structure after 10 cycles catalysis (Supplementary Fig. S19), all confirming the stability of the PCP framework during the catalytic reactions. Meanwhile, at the end of the reaction, inductively coupled plasma analysis of the reaction mixture filtrate revealed no Zn leaching, indicating that the catalytic reaction is indeed heterogeneous in nature.

Reply to the Reviewer #3's comments

This report synthesized a crystalline porous structure called Zn-DPA, using two ligands of H₃tca and dpa, which was characterized to be interpenetrated and exhibited a high affinity to CO₂, as determined by adsorption isotherms and single-crystal XRD, and demonstrated the catalysis results with different sized substrates. The following questions should be considered in revision of the paper.

1. The authors claimed that the flexibility and rigidity of the interpenetrated frameworks are responsible for the CO₂ trapping and size-selectivity in catalysis, respectively. Can the authors give direct evidence to support this point? Or what if using H₃tca and dpa as ligand separately to synthesize PCPs?

Response: Thanks for the kind question. Direct observation of gas molecules confined in the nanospace using X-ray or neutron diffraction technique is quite

valuable, because it leads to deep insight into the adsorption mechanism and the actual state of the adsorbate at a molecular level. However, so far, reports of crystal structures containing adsorbed gas molecules have been still limited in number because in general it is not necessary for gas molecules in the nanospace to form a periodic structure that is commensurate with the electrostatic field formed in the pore, especially in the case of a rigid host. When the host framework is very rigid and there is no specific interaction site, the guest molecules tend to form the densest structure, that is to say, the periodicity of the framework does not correspond to that of the assembled structure of guest molecules. In that case, we cannot find the exact position of the guest molecules so that the crystal structure of a PCP with guest molecules cannot be solved. A limited number of reports revealed that commensurate CO₂-trapping crystal structures were produced when the host has sufficient flexibility to trap guest molecules, resulting in a high probability of developing gas-trapping structures. (e.g. Vaidhyanathan, R. *et al. Science* **330**, 650–653 (2010); Warren, J. E., *et al. Angew. Chem. Int. Ed.* **53**, 4592–4596 (2014); Ma, Y. S., *et al. J. Am. Chem. Soc.* **137**, 15825–15832 (2015).)

For demonstrating that the rigidity of our PCP is responsible for size-selectivity in catalysis, the cycloaddition reaction of carbon dioxide and epoxides was catalyzed by a flexible PCP, MOF-508b (Zn(BDC)(4,4'-Bipy)_{0.5}, BDC = 1,4-benzenedicarboxylic acid, 4,4'-Bipy=4,4'-bipyridine) under the same conditions as that catalyzed by Zn-DPA. As shown in Table S7, the reaction yields catalyzed by MOF-508a from related epoxides are 42% for 2-(phenoxy)methyl oxirane, 47% for allyl glycidyl ether, 45% for butylglycidyl ether, 46% for glycol diglycidyl ether, 44% for glycidyl methacrylate, 42% for 2-((4-nitrophenoxy)methyl)oxirane, 43% for 2-((4-methoxyphenoxy)methyl) oxirane, 41% for 1,3-bis(oxiran-2-ylmethoxy)-benzene and 42% for bis(4-(oxiran-2-ylmethoxy)phenyl)methane with corresponding turnover frequency (TOF) values of 1115, 1250, 1195, 1225, 1170, 1115, 1145, 1090, and 1115 h⁻¹ per paddlewheel Zn₂ cluster. No significant difference in the conversion was observed even when bulky epoxides such as 1,3-bis(oxiran-2-ylmethoxy)benzene

and bis(4-(oxiran-2-ylmethoxy)phenyl)methane were used as the reaction substrates. The comparison of the yields for all ten products clearly reveals that our PCP exhibits the size selectivity to small and large substrates in the reactions although the these two catalysts are both two-fold interpenetrated frameworks containing open metal sites, which highlights the importance of the rigidity character of the channels for the size-selective catalysis for CO₂ fixation.

Supplementary Table S7. MOF-508b-catalyzed coupling of epoxides with CO₂.^a

Entry	Epoxides	Yield ^b (%)	TON ^c	TOF ^d
1		42	2230	1115
3		47	2500	1250
4		45	2390	1195
5		46	2450	1225
6		44	2340	1170
7		42	2230	1115
8		43	2290	1145
9		41	2180	1090
10		42	2230	1115

^a Reaction conditions: epoxide (20 mmol), catalyst (3.75 μmol, based on Zn₂ cluster) and TBABr (0.3 mmol) under carbon dioxide (1 MPa), 373 K and 2 h. ^b Yield of isolated product was determined from by ¹H NMR spectroscopy. ^c Moles of cyclic carbonate per mole of catalyst MOF-508b. ^d Moles of cyclic carbonate per mole of catalyst MOF-508b per hour.

2. *The molecular size of the substrates used should be compared with the channels of the PCP catalysts.*

Response: Thanks for the kind suggestion. The molecular size of selected substrates are added in Table 1 of the revised manuscript. The substrates chosen for the reaction were 2-(phenoxy)methyl)oxirane (molecular size, $4.2 \times 9.4 \text{ \AA}^2$), allyl glycidyl ether (molecular size, $3.1 \times 8.2 \text{ \AA}^2$), butylglycidyl ether (molecular size, $3.1 \times 9.9 \text{ \AA}^2$), glycidyl methacrylate (molecular size, $4.4 \times 8.6 \text{ \AA}^2$), glycol diglycidyl ether (molecular size, $3.1 \times 10.9 \text{ \AA}^2$), 2-((4-nitrophenoxy)methyl)oxirane (molecular size, $4.2 \times 10.2 \text{ \AA}^2$), 2-((4-methoxyphenoxy)methyl)oxirane (molecular size, $4.2 \times 11.6 \text{ \AA}^2$), 1,3-bis(oxiran-2-ylmethoxy)benzene (molecular size, $5.7 \times 12.0 \text{ \AA}^2$) and bis(4-(oxiran-2-ylmethoxy)phenyl)methane (molecular size, $6.0 \times 16.5 \text{ \AA}^2$). Structural simulations of the substrates demonstrated that the three-dimensionally running channels with dimensions of $5.8 \times 11.5 \text{ \AA}^2$ are available for the first eight small substrates accommodation and exchange, whereas not large enough to allow 1,3-bis(oxiran-2-ylmethoxy)benzene and bis(4-(oxiran-2-ylmethoxy)phenyl)methane molecules in and out. This is also in accord with the catalytic efficiency.

3. *Since many other interpenetrated PCPs already exist in the literature, what are the unique features of this proposed Zn-DPA endowing their properties in fixing CO₂? The authors should provide more insightful discussions on the design rationales.*

Response: We highly appreciate the reviewer's kind questions. The advantages of our proposed Zn-DPA in fixing CO₂ are the following. (1) well-orientation of Lewis acidic open sites toward the channels, facilitating the access of CO₂ and substrate molecules to open zinc sites to activate them. These interactions are confirmed by the CO₂-loaded crystal structure and theoretical calculation for the binding energy of CO₂ and epoxide substrates in this framework that we supplemented. And we added the information about Lewis acidic centers in page 5, line 13: "*It is noteworthy that two unsaturated zinc centers on both ends of a trinuclear unit are well-oriented toward*

the channels, facilitating their full accessibility for the substrates to their open sites of Lewis acidic centers.”

(2) the maintenance of its channels even upon desolvation, ensuring to impose size- and shape-selective restrictions for substrates during the catalytic process. It is known that interpenetrated frameworks often show kinetically non-porous nature after the removal of solvents. Such catalytic processes often originate from surface catalysis of bare metal sites. This is also demonstrated by a compared catalyst, MOF-508a ($\text{Zn}(\text{BDC})(4,4'\text{-Bipy})_{0.5}$), in which a dense doubly interpenetrating 3D framework is formed upon desolvation. We found that no significant difference in the conversion was observed when small epoxides and bulky epoxides were respectively used as the reaction substrates. Although our PCP and MOF-508b are both two-fold interpenetrated frameworks containing open metal sites, they exhibit different catalytic effect, highlighting the importance of the rigidity character of the channels in a interpenetrated framework for the size-selective catalysis for CO_2 fixation. Thus, the dual structural attributes—flexibility and rigidity in a PCP material is vital for CO_2 capture, activation and size-selective conversion. We believe our strategy will open up a new dimension of porous compounds as platforms for determining the gas-loaded crystal structures and efficient utilization of C1 resources.

Reviewers' comments:

Reviewer #1 (Remarks to the Author):

In general, this is a nice paper describing the reactivity of a lewis acid sites in a MOF, subsequently used for cycloaddition reactions with size-selective reactivity. The in-situ XRD work at the start, however is poorly described, and in my opinion the refinement of the '100%' occupied CO₂ sites needs re-visited. This is especially important as so little in-situ X-ray diffraction data showing CO₂ uptake is actually available. For example, one of the CO₂ molecules has a <OCO angle of 150 degrees. This is clearly wrong.

My other main complaint is the terminology. A framework cannot be flexible and rigid. Its an oxymoron that should not be applied to a framework. Even a small amount of flexibility can result in unusual materials properties, but some flexibility is still needed. To claim that a MOF is rigid, whose definition is 'unable to bend or be forced out of shape; not flexible.' Is a wrong use of the word in this context.

1. Page 4 line 11.

Flexibility and rigidity at the same time is impossible. I understand what the authors are trying to do, but I would accommodate this as a flexible framework with the resulting channels a perfect fit for the desired sorbed species.

2. Page 4 line 19, how can the framework change, with no resulting change in channel size, at all.

3. Page 8 line 4/5. Is the CO₂ not solid at 123K?

4. Page 8 line 13. This is a throw away statement. Why are the dihedral angles important?

5. Page 8 line 14. Are the CO₂ molecules occupied at 100%? It would seem odd to be specifying C=O bond lengths and angles for an occupied CO₂ molecule with large thermal motion that quite frankly should be restrained. One of the CO₂ molecules has a bent angle. Unless this is a new form of CO₂, or formate, the authors need to look at the data much more closely.

6. Page 8 line 15, you can state clearly that the occupancy is 100%? Why is the thermal motion so high then? With such a large correlation between the occupancy and thermal motion, I find this hard to believe.

7. Point (1) above. Page 9 line 4/5, really can you say this? No change at all? Whats the RMS deviation of the framework before and after? Or the RMS deviation difference between the pore space before and after. If your going to make a claim like this, and make it a selling point of the paper, it needs to be quantified.

8. Page 12, line 3 is this really a rigid framework? How to you quantify 'rigidity character'. And is 'rigidity' a word that should be used in this context? Again I do not believe so.

Reviewer #2 (Remarks to the Author):

The authores have addressed my concerns. I would like to recommend it for publication in Nature Communications.

Reviewer #3 (Remarks to the Author):

Since the title claims "direct observation", and the authors claimed that the flexibility and rigidity of the interpenetrated frameworks are responsible for the CO₂ trapping and size-selectivity in catalysis, respectively. I still prefer that the authors could give more direct evidences to support these points.

Point-by-point response to the referees' comments

The reviewers' comments are written with *Italic font* and replies are written in **blue**.

Reply to the Reviewer #1's comments

In general, this is a nice paper describing the reactivity of a lewis acid sites in a MOF, subsequently used for cycloaddition reactions with size-selective reactivity. The in-situ XRD work at the start, however is poorly described, and in my opinion the refinement of the '100%' occupied CO₂ sites needs re-visited. This is especially important as so little in-situ X-ray diffraction data showing CO₂ uptake is actually available. For example, one of the CO₂ molecules has a <OCO angle of 150 degrees. This is clearly wrong.

My other main complaint is the terminology. A framework cannot be flexible and rigid. Its an oxymoron that should not be applied to a framework. Even a small amount of flexibility can result in unusual materials properties, but some flexibility is still needed. To claim that a MOF is rigid, whose definition is 'unable to bend or be forced out of shape; not flexible.' Is a wrong use of the word in this context.

1. Page 4 line 11.

Flexibility and rigidity at the same time is impossible. I understand what the authors are trying to do, but I would accommodate this as a flexible framework with the resulting channels a perfect fit for the desired sorbed species.

Response: As pointed out by the reviewer, flexibility and rigidity simultaneously in a framework makes a mistake in using. What we meant is co-existence of “local flexibility” and “global rigidity” in our compound; a framework motif itself is rigid but a ligand as building block has a flexible part, and therefore “adaptable channels” in an entangled framework could be realized. Accordingly, we revised our manuscript as follows.

Page 2, line 4: *“Here, we present a dynamic porous coordination polymer (PCP) material with local flexibility, in which the propeller-like ligands rotate to permit CO₂ trapping.”*

Page 4, line 9: *“Thus our aim is to introduce local flexibility so effectively that CO₂ are captured by size and shape-induced fit and also that all the reactants still possess degree of freedom for the coupling reaction.”*

Page 13, line 18: *“The key to success is to introduce a durable interpenetrated framework, a local flexibility to induce adjustable capture and catalytic Lewis acid sites in the pores.”*

2. Page 4 line 19, how can the framework change, with no resulting change in channel size, at all.

Response: Upon desolvation and CO₂ uptake, three phenyl rings of 4,4',4''-tricarboxyltriphenylamine ligands in PCPs show slight rotational motion. The solvent-accessible volume also changed from original 1,858.3 Å³ (31.2% of the unit-cell volume) for Zn-DPA·2H₂O into 1807.9 Å³ (30.5%) for Zn-DPA and 1839.3 Å³ (30.9%, when CO₂ molecules are omitted) for Zn-DPA·1.21CO₂, respectively. The corresponding channels with dimensions of 5.8 × 11.5 Å² changed to 5.7 × 11.0 Å² (Zn-DPA) and 6.0 × 11.3 Å² (Zn-DPA·1.21CO₂). We have carefully revised the related expressions.

Page 4 line 19: *“Their propeller-like ligands 4,4',4''- tricarboxyltriphenylamine undergo rotational rearrangement in response to the removal and rebinding of guest molecules, resulting in slight changes of their channels.”*

3. Page 8 line 4/5. Is the CO₂ not solid at 123K?

Response: Sorry to make confusion on the detailed information about CO₂-loaded single crystal collection. 95 kPa CO₂ was loaded into evacuated crystals of Zn-DPA in the capillary at 293 K, then the capillary was sealed. The capillary sample was slowly cooled down to 123 K, in which CO₂ gas is absorbed into the single crystal, then subjected to the SXRD analysis. Because the amount of CO₂ gas in the

sealed-capillary was limited, it may not be sufficient to fill all adsorption sites in Zn-DPA with full occupancy. In fact, 1.21 CO₂ per formula unit was observed by SXRD analysis although the full occupancy of CO₂ determined by gas sorption experiments is 2.

4. Page 8 line 13. This is a throw away statement. Why are the dihedral angles important?

Response: Thanks for the reviewer's kind suggestion. As shown in the Table 1 below, only little changes have occurred for the dihedral angles in our PCPs. Thus, the description about dihedral angles on Page 8 line 13 was deleted.

Table 1. Comparison of crystal structures between Zn-DPA·2H₂O, Zn-DPA and Zn-DPA·1.21CO₂.

PCP	dihedral angles between the planes (deg)				
	Zn-O-Zn vs phenyl ^a	Zn-O-Zn vs pyridine ^b	Zn-O-Zn vs O-C-O ^c	O-C-O vs phenyl ^d	phenyl vs pyridine ^e
Zn-DPA·2H ₂ O	13.53	87.44	12.23	2.68	88.15
Zn-DPA	13.29	88.91	12.67	4.17	89.48
Zn-DPA·1.21CO ₂	12.65	88.85	13.10	3.87	85.81

^a Phenyl rings involving Zn1, Zn2, O2, and O3. ^b Pyridine rings involving Zn1, Zn2, O2, and O3. ^c The plane including Zn1, Zn2, O2, and O3 vs the plane including O2, C1 and O3. ^d Phenyl rings involving O2, C1 and O3. ^e O1-Zn1-O3-C1. ^f O1-Zn2- O2-C1.

5. Page 8 line 14. Are the CO₂ molecules occupied at 100%? It would seem odd to be specifying C=O bond lengths and angles for an occupied CO₂ molecule with large thermal motion that quite frankly should be restrained. One of the CO₂ molecules has a bent angle. Unless this is a new form of CO₂, or formate, the authors need to look at the data much more closely.

6. Page 8 line 15, you can state clearly that the occupancy is 100%? Why is the thermal motion so high then? With such a large correlation between the occupancy

and thermal motion, I find this hard to believe.

Response: We have re-resolved the crystal data to make occupied CO₂ molecules more reasonable, including their thermal motion, C=O bond lengths and angles. The CO₂ occupancies were determined by free structural refinement to be 0.760(78), 0.211(68) and 0.238(37) at sites I, II and III, respectively. During refinement, for CO₂-II and CO₂-III, the C=O bond distances were fixed at 1.15 Å, and the O-O distances were also fixed at 2.30 Å. The geometries of these three CO₂ molecules appear with angles [\angle O-C-O: CO₂-I, 176.6(2)°; CO₂-II, 174.9(3)°; CO₂-III, 176.8(3)°] and the C-O bond lengths (CO₂-I, 1.12(1), 1.16(6) Å; CO₂-II, 1.15(1), 1.15(1) Å; CO₂-III, 1.15(0), 1.15(0) Å). In fact, we revised the relative information as follows.

Page 8, line 13: *“It is significant that the CO₂ geometries [C–O bond lengths: CO₂-I, 1.12(1), 1.16(6) Å; CO₂-II, 1.15(1), 1.15(1) Å; CO₂-III, 1.15(0), 1.15(0) Å; \angle O–C–O, CO₂-I, 176.6(2)°; CO₂-II, 174.9(3)°; CO₂-III, 176.8(3)°] are similar to that of solid CO₂.²⁹”*

The new CIF file and checkCIF report are uploaded.

7. Point (1) above. Page 9 line 4/5, really can you say this? No change at all? Whats the RMS deviation of the framework before and after? Or the RMS deviation difference between the pore space before and after. If your going to make a claim like this, and make it a selling point of the paper; it needs to be quantified.

Response: We appreciate the reviewer for pointing out this problem. Upon CO₂ adsorption, the rotational rearrangements of tca³⁻ ligands caused small framework deformations, the distance between the different interpenetrated nets increased slightly from 17.058 Å in Zn-DPA to 17.080 Å in Zn-DPA·1.21CO₂; The solvent-accessible volume also changed from 1807.9 Å³ (30.5%) for Zn-DPA to 1839.3 Å³ (30.9%, when CO₂ molecules are omitted) for Zn-DPA·1.21CO₂. It is precisely because the main framework has sufficient flexibility to trap CO₂ molecules to develop gas-trapping structures. On the other hand, the adaptable channels of the PCP is responsible for size-selectivity in catalysis simultaneously. Accordingly, we revised the sentences in Page 8 line 24/25: *“The distance between the different*

interpenetrated nets is 17.070 Å and the void volume is 30.9% (when CO₂ molecules are omitted) upon CO₂ adsorption (Figure 1h). ”

8. *Page 12, line 3 is this really a rigid framework? How to you quantify ‘rigidity character’. And is ‘rigidity’ a word that should be used in this context? Again I do not believe so.*

Response: According to the reviewer’s comments, we revised the section “*Notably, the catalytic effect in a open-dense flexible PCP was not different for small and bulky epoxide substrates, which highlights the importance of the adaptable channels for the size-selective catalysis for CO₂ fixation (Supplementary Table S7).*” in the main text.

Reply to the Reviewer #2’s comments

The authores have addressed my concerns. I would like to recommend it for publication in Nature Communications.

Response: We appreciate the reviewer’s inputs and understanding our efforts to improve our paper.

Reply to the Reviewer #3’s comments

Since the title claims "direct observation", and the authors claimed that the flexibility and rigidity of the interpenetrated frameworks are responsible for the CO₂ trapping and size-selectivity in catalysis, respectively. I still prefer that the authors could give more direct evidences to support these points.

Response: We really appreciate the reviewer’s suggestion. Actually, we are able to locate the position and clarify the interaction of CO₂ with the framework through CO₂-loaded single crystal, while the catalytic reaction process is not supported by the direct single crystal data. Even so, the interaction between the framework and the substrate epoxides have been demonstrated by FT-IR spectra, ¹H NMR data and density functional theory calculation. The use of “direct observation” in the title is indeed inappropriate, thus, we revised the title into “*Carbon Dioxide Capture and Efficient Fixation in a Dynamic Porous Coordination Polymer*”.

Reviewers' comments:

Reviewer #4 (Remarks to the Author):

Crystallographic Review

Three structure determinations of the same Zn MOF are presented: one solvated with water, one evacuated and one loaded with CO₂. Significant revisions and clarifications are needed before these structures can be recommended for publication.

The authors are to be commended for their efforts to carry out this challenging experiment and their positive response to previous referee comments. However, there are still deficiencies in the refinement and analysis that mean these structures do not support the weight of discussion made about them in the manuscript.

General Notes

Table S3 needs ESDs. The ESDs on all intermolecular distances should also be calculated by means of a full least squares matrix refinement using an RTAB DIST instruction (if using ShelXL) rather than directly calculating from the coordinated and ESDs in the CIF alone.

All structures should make use of the `_refine_special_details` field in the CIF to give a full and detailed account of all restraints and constraints used in the refinement. This information should be reproduced in the SI.

CIF validation response forms (VRF) should be included for all Alert A and Alert B notifications in the CIF report explaining the origin of any unusual results.

Given the non-ambient conditions under which some of the structures are measured it would be helpful to full accounts of the crystal preparation and mounting methods in the following fields:

`_diffrn_measurement_specimen_support`

`_diffrn_ambient_environment`

`_diffrn_crystal_treatment`

All of the provided CIFs give generic descriptions of the instruments used. The manuscript describes use of a Rigaku Saturn 70 CCD, however the software used for data processing is listed as Bruker – is this correct?

The authors have used the restraint DELU in some structures to restrain the anisotropic displacement parameters – this restraint has been superseded by the new restraint RIGU in the latest versions of SHELXL.

ZN DPA H₂O

The temperature reported in the table is -123 K however the refinement file has the instruction TEMP 23. This should be -150 (units °C) – fixed hydrogen geometries depend on this being correct.

Omit 0 50: justify use of this instruction in the experimental

Explain use of DFIX 3.0 restraint on o1w o2w? Description of purpose of all geometric restraints is critical given the detailed discussion of the framework geometry.

Large electron density peak close to disordered water: Q1 1.08 eA⁻³ 1.15 Å from O2W (occ 025 fix) – could it be another disorder component?

O1W' and O2W' both in disorder PART 2 but 0.832 Å apart – they should be in different PARTs.

The refinement of the disordered water molecules is poorly handled. Their ellipsoid have extremely large variation of Ueq values (0.073 to 0.300) whilst their occupancies have been fixed at nominal values of 0.25 and 0.5. This is physically implausible. A better strategy in this situation would be to fix or restrain their isotropic displacement parameters to a sensible value and freely refine their occupancies.

ZN DPA EMPTY

How was crystal transferred from vacuum to diffractometer to minimise resolution? Given the importance to the discussion of proving that the structure is completely evacuated it would prudent for the authors to report and discuss the Squeeze output statistics for both the solvated and evacuated crystals. I have run a Squeeze on the solvated structure (with the water omitted) which indicates that the evacuated crystal contains around 25% of the pore electron count of the solvated one. Given the importance of fully evacuating prior to back filling with CO₂ these

numbers should be reported and examined in detail.
Centre of gravity not in cell Alert – Fix.

Reported Squeeze

i.e. Centre of Gravity, Solvent Accessible Volume,
Recovered number of Electrons in the Void and
Details about the Squeezed Material

```
loop_  
_platon_squeeze_void_nr  
_platon_squeeze_void_average_x  
_platon_squeeze_void_average_y  
_platon_squeeze_void_average_z  
_platon_squeeze_void_volume  
_platon_squeeze_void_count_electrons  
_platon_squeeze_void_content  
1 0.750 0.010 0.250 440 25 ' '  
2 0.250 0.017 0.250 440 25 ' '  
3 0.750 -0.042 0.750 438 25 ' '  
4 0.250 -0.048 0.750 438 25 ' '  
5 0.000 0.195 0.250 13 0 ' '  
6 0.500 0.304 0.750 13 0 ' '  
7 0.500 0.695 0.250 13 0 ' '  
8 1.000 0.804 0.750 13 0 ' '
```

Author Squeeze output for H2O Structure

SQUEEZE RESULTS (Version = 260918)
Note: Data are Listed for all Voids in the P1 Unit Cell
i.e. Centre of Gravity, Solvent Accessible Volume,
Recovered number of Electrons in the Void and
Details about the Squeezed Material

```
loop_  
_platon_squeeze_void_nr  
_platon_squeeze_void_average_x  
_platon_squeeze_void_average_y  
_platon_squeeze_void_average_z  
_platon_squeeze_void_volume  
_platon_squeeze_void_count_electrons  
_platon_squeeze_void_content  
1 0.750 0.009 0.250 453 90 ' '  
2 0.250 0.014 0.250 453 90 ' '  
3 0.750 -0.041 0.750 450 90 ' '  
4 0.250 -0.045 0.750 450 90 ' '  
5 0.000 0.204 0.250 14 0 ' '  
6 0.500 0.296 0.750 14 0 ' '  
7 0.500 0.704 0.250 14 0 ' '  
8 1.000 0.796 0.750 14 0 ' '  
_platon_squeeze_void_probe_radius 1.20  
_platon_squeeze_details ?
```

ZN DPA CO2

In a structure of this complexity I would expect a large amount of detail about the refinement in the `_refine_special_details` section.

Logical naming of CO2 residues would help interpretation.

Authors should explain that the CO2 occupancies have been fixed globally per molecule – do all the atoms refine to similar occupancies if refined freely and independently?

The same esd has been used on DFIX 1,2 and 1,3 restraints (0.001). the 1,3 bonds restraints ESDs should be larger than those of 1,2 bonds.

TEMP instruction missing.

Efficacy of outgassing procedure is not reported. The crystal loaded with CO₂ does not seem to be the same crystal that was used to determine the 'outgas' empty structure. Can you be sure that under these conditions (shorter outgassing time) the crystal is fully desolvated. It can be much harder to outgas mounted single crystals under vacuum due to poor thermal conduction.

The refinement of CO₂-I is good – there is compelling evidence it is there and the refinement is appropriate and plausible.

Sites II and III are problematic. The ISOR restraint applied to their ADP makes them to all purposes isotropic – refining them as isotropic makes no difference to the R1 value of the structure. Sites II and III have much larger Ueq values than site I. This calls into question their refined occupancies of 0.21 and 0.25. If they are refined with isotropic parameters fixed as plausible values of 0.15 then their occupancies drop to 0.15. At this point I would dispute whether such low occupancy residues can be confidently identified.

The reason for the use of a DAMP (1000) instruction in the final refinement job should be discussed/reported. If it is to aid convergence of the low occupancy CO₂ residues then it further calls into question whether they can be adequately modelled with this data. The structure is very poorly converging – it took 12 repeats of 25 cycles of least squares to converge. This is further evidence of a poor model for the data.

The manuscript describes interactions between adjacent CO₂ moieties. Firstly, their low partial occupancies mean that it is not certain that they ever occupy the same unit cell simultaneously. The very large ADP values and low occupancies of sites II and III mean that geometric interactions can only be described in very broad terms. In my opinion Site I is mutually exclusive to site II: the C...O distance of 2.33 Å is very short. The sum of the VdW radii for these atoms is 3.20 and intermolecular C...O distances in solid CO₂ are 3.11 Å. Benchmark DFT calculations for C...O T-interactions give values of around ca 2.8 Å. Such a short reported distance is not supported by this weak data.

Point-by-point response to the referees' comments

The reviewers' comments are written with *Italic font* and replies are written in blue.

Reply to the Reviewer #4's comments

Crystallographic Review

Three structure determinations of the same Zn MOF are presented: one solvated with water, one evacuated and one loaded with CO₂. Significant revisions and clarifications are needed before these structures can be recommended for publication. The authors are to be commended for their efforts to carry out this challenging experiment and their positive response to previous referee comments. However, there are still deficiencies in the refinement and analysis that mean these structures do not support the weight of discussion made about them in the manuscript.

Response: We thank the reviewer for his/her constructive comments and advices on our crystallographic data. Following the reviewer's advice, we carried out the data collection and their refinements again to improve all our crystallographic data.

We repeated many experiments in obtaining fine single-crystal structures of more CO₂ sorption complexes Zn-DPA@CO₂. A single crystal was selected to put in a capillary and evacuated at 120 °C under reduced pressure (below 10⁻² Pa) overnight. Then, CO₂ was slowly introduced into the capillary until the pressure reaching 101 kPa at 195 K. After 10 min at 195 K under 101 kPa CO₂ atmosphere, the glass capillary was sealed using small torch flame. The gas loaded crystals were mounted onto a Rigaku XtaLAB AFC10 diffractometer equipped with Mo K α ($\lambda = 0.71073 \text{ \AA}$) optic and slowly cooled down and diffraction data were collected at 183 K, realizing that CO₂ content found by SCXRD-result became largest (two CO₂ molecules). So, we also re-measured Zn-DPA-empty and as-synthesized Zn-DPA at 183 K. The 183-K data set yielded the better refinement parameters [refinement factor (R) = 4.92%, weighted R (Rw) = 14.77% for as-synthesized Zn-DPA, (R) = 4.74%, weighted R (Rw) = 10.15% for Zn-DPA-empty, (R) = 4.18%, weighted R (Rw) =

11.82% for Zn-DPA@CO₂] compared to the previous data and will be used for structural discussions and resolve the related deficiencies in the refinement and analysis that the reviewer pointed out below.

The new CIF files and checkCIF reports are uploaded.

General Notes

Table S3 needs ESDs. The ESDs on all intermolecular distances should also be calculated by means of a full least squares matrix refinement using an RTAB DIST instruction (if using ShelXL) rather than directly calculating from the coordinated and ESDs in the CIF alone.

Response: According to the reviewer's suggestion, we have used an RTAB DIST instruction to calculate the ESDs on all intermolecular distances. The calculation results are listed as follows:

Zn-DPA·2H₂O

NI -	Distance	Angles
C12	1.4140 (0.0032)	
C5	1.4268 (0.0034)	117.37 (0.22)
C19	1.4283 (0.0033)	118.28 (0.21) 118.28 (0.21)

Zn-DPA

NI -	Distance	Angles
C19	1.4153 (0.0028)	
C12	1.4302 (0.0029)	117.87 (0.18)
C5	1.4316 (0.0029)	117.31 (0.19) 117.88 (0.18)

Zn-DPA·2CO₂

NI -	Distance	Angles
C12	1.4123 (0.0025)	
C19	1.4201 (0.0027)	119.09 (0.16)
C5	1.4315 (0.0025)	116.49 (0.18) 119.00 (0.17) ”.

We have added the above ESDs in Table S3 in the revised supporting information.

All structures should make use of the `_refine_special_details` field in the CIF to give a full and detailed account of all restraints and constraints used in the refinement. This information should be reproduced in the SI.

Response: More details regarding the restraints and constraints have been added in the new CIFs and the revised Supplementary Information.

CIF validation response forms (VRF) should be included for all Alert A and Alert B notifications in the CIF report explaining the origin of any unusual results.

Response: All Alert A and Alert B have been explained in the revised Supplementary Information.

Given the non-ambient conditions under which some of the structures are measured it would be helpful to full accounts of the crystal preparation and mounting methods in the following fields:

`_diffrn_measurement_specimen_support`

`_diffrn_ambient_environment`

`_diffrn_crystal_treatment`

Response: The details regarding the crystal preparation and mounting methods have been added in the corresponding fields of the new CIFs.

All of the provided CIFs give generic descriptions of the instruments used. The manuscript describes use of a Rigaku Saturn 70 CCD, however the software used for data processing is listed as Bruker – is this correct?

Response: Sorry to make confusion on the descriptions of the instruments used. The measurement device type was corrected as “Rigaku XtaLAB AFC10” in the new CIFs.

The authors have used the restraint DELU in some structures to restrain the anisotropic displacement parameters – this restraint has been superseded by the new

restraint RIGU in the latest versions of SHELXL.

Response: The restraint DELU was not used in the new crystal data.

ZN DPA H2O

The temperature reported in the table is -123 K however the refinement file has the instruction TEMP 23. This should be -150 (units °C) – fixed hydrogen geometries depend on this being correct.

Response: The new diffraction data for these three crystals as-synthesized Zn-DPA Zn-DPA-empty and Zn-DPA@CO₂ were all collected at 183 K.

Omit 0 50: justify use of this instruction in the experimental

Explain use of DFIX 3.0 restraint on o1w o2w? Description of purpose of all geometric restraints is critical given the detailed discussion of the framework geometry.

Response: “Omit 0 50” and “DFIX 3.0 restraint on o1w o2w” were not used in the new crystal data.

Large electron density peak close to disordered water: Q1 1.08 eA-3 1.15 Å from O2W (occ 025 fix) – could it be another disorder component?

O1W' and O2W' both in disorder PART 2 but 0.832 Å apart – they should be in different PARTs.

The refinement of the disordered water molecules is poorly handled. Their ellipsoid have extremely large variation of Ueq values (0.073 to 0.300) whilst their occupancies have been fixed at nominal values of 0.25 and 0.5. This is physically implausible. A better strategy in this situation would be to fix or restrain their isotropic displacement parameters to a sensible value and freely refine their occupancies.

Response: According to the reviewer’s comments, we have refined the new crystal data to make the disordered water molecules more reasonable.

ZN DPA EMPTY

How was crystal transferred from vacuum to diffractometer to minimise resolution?

Given the importance to the discussion of proving that the structure is completely evacuated it would be prudent for the authors to report and discuss the Squeeze output statistics for both the solvated and evacuated crystals. I have run a Squeeze on the solvated structure (with the water omitted) which indicates that the evacuated crystal contains around 25% of the pore electron count of the solvated one. Given the importance of fully evacuating prior to back filling with CO₂ these numbers should be reported and examined in detail.

Centre of gravity not in cell Alert – Fix.

Reported Squeeze

i.e. Centre of Gravity, Solvent Accessible Volume,

Recovered number of Electrons in the Void and

Details about the Squeezed Material

loop_

_platon_squeeze_void_nr

_platon_squeeze_void_average_x

_platon_squeeze_void_average_y

_platon_squeeze_void_average_z

_platon_squeeze_void_volume

_platon_squeeze_void_count_electrons

_platon_squeeze_void_content

1 0.750 0.010 0.250 440 25 ''

2 0.250 0.017 0.250 440 25 ''

3 0.750 -0.042 0.750 438 25 ''

4 0.250 -0.048 0.750 438 25 ''

5 0.000 0.195 0.250 13 0 ''

6 0.500 0.304 0.750 13 0 ''

7 0.500 0.695 0.250 13 0 ''

8 1.000 0.804 0.750 13 0 ''

Author Squeeze output for H2O Structure

SQUEEZE RESULTS (Version = 260918)

Note: Data are Listed for all Voids in the P1 Unit Cell

i.e. Centre of Gravity, Solvent Accessible Volume,

Recovered number of Electrons in the Void and

Details about the Squeezed Material

loop_

_platon_squeeze_void_nr

_platon_squeeze_void_average_x

_platon_squeeze_void_average_y

_platon_squeeze_void_average_z

_platon_squeeze_void_volume

_platon_squeeze_void_count_electrons

_platon_squeeze_void_content

1 0.750 0.009 0.250 453 90 ''

2 0.250 0.014 0.250 453 90 ''

3 0.750 -0.041 0.750 450 90 ''

4 0.250 -0.045 0.750 450 90 ''

5 0.000 0.204 0.250 14 0 ''

6 0.500 0.296 0.750 14 0 ''

7 0.500 0.704 0.250 14 0 ''

8 1.000 0.796 0.750 14 0 ''

_platon_squeeze_void_probe_radius 1.20

_platon_squeeze_details ?

Response: Thanks for the reviewer's kind suggestion. The detailed information for crystal transferred from vacuum to diffractometer is as follows: As-synthesized single crystals were air-dried. One single crystal suitable for the diffraction measurement was put into a glass capillary. The capillary was connected to handmade gas pressure

handling unit, and was evacuated (below 10^{-2} Pa) at 120 °C overnight. Then, the capillary was sealed using small torch flame with keeping vacuum condition inside the capillary. The sealed capillary was mount on the diffractometer at 183 K.

In the new crystal data, application of procedure SQUEEZE (program PLATON) did not bring about a significant improve of refinement results and therefore was not retained for the final refinement. For as-synthesized Zn-DPA alert B: Structure Contains Solvent Accessible VOIDS of . 137 Ang**3 occurred. For Zn-DPA-empty, alert A: VERY LARGE Solvent Accessible VOID(S) in Structure ! Info occurred. They might be due to the existence of a large solvent channel in the structure itself. All Alert A and Alert B have been explained in the revised Supplementary Information.

TGA also clearly indicated that the guest water molecules were completely excluded in this condition (Supplementary Fig. S8). In addition, based on the new crystal data, the Squeeze results are also listed as follows:

Zn-DPA-2H2O

```
# SQUEEZE RESULTS (Version = 241018)
# Note: Data are Listed for all Voids in the P1 Unit Cell
# i.e. Centre of Gravity, Solvent Accessible Volume,
# Recovered number of Electrons in the Void and
# Details about the Squeezed Material
loop_
  _platon_squeeze_void_nr
  _platon_squeeze_void_average_x
  _platon_squeeze_void_average_y
  _platon_squeeze_void_average_z
  _platon_squeeze_void_volume
  _platon_squeeze_void_count_electrons
  _platon_squeeze_void_content
  1 0.250 0.097 0.135 915 321 ''
```

2	0.750	-0.047	0.177	915	321 ''
3	0.000	0.045	0.250	13	-1 ''
4	0.500	0.455	0.750	13	-1 ''
5	0.500	0.545	0.250	13	-1 ''
6	1.000	0.955	0.750	13	-1 ''

Zn-DPA EMPTY

SQUEEZE RESULTS (Version = 241018)

Note: Data are Listed for all Voids in the P1 Unit Cell

i.e. Centre of Gravity, Solvent Accessible Volume,

Recovered number of Electrons in the Void and

Details about the Squeezed Material

loop_

 _platon_squeeze_void_nr

 _platon_squeeze_void_average_x

 _platon_squeeze_void_average_y

 _platon_squeeze_void_average_z

 _platon_squeeze_void_volume

 _platon_squeeze_void_count_electrons

 _platon_squeeze_void_content

1	0.250	-0.052	0.250	443	25 ''
2	0.750	0.052	0.250	443	25 ''
3	0.250	0.021	0.750	441	25 ''
4	0.750	-0.070	0.750	441	25 ''
5	0.500	0.058	0.250	11	0 ''
6	0.541	0.041	0.700	8	0 ''
7	0.459	0.041	0.800	8	0 ''
8	0.000	0.442	0.750	11	0 ''
9	0.041	0.458	0.200	8	0 ''
10	0.959	0.458	0.300	8	0 ''

11	0.000	0.560	0.250	10	0''
12	0.042	0.543	0.699	7	0''
13	0.958	0.543	0.801	7	0''
14	0.500	0.940	0.750	10	0''
15	0.542	0.957	0.199	7	0''
16	0.458	0.957	0.301	7	0''

ZN DPA CO2

In a structure of this complexity I would expect a large amount of detail about the refinement in the `_refine_special_details` section.

Response: The detail about the refinement have been added in the new CIF.

Logical naming of CO2 residues would help interpretation.

Response: As pointed out by the reviewer, CO₂ residues were named as O(1A)-C(1A)-O(2A), O(1B)-C(1B)-O(2B), O(1C)-C(1C)-O(2C) and O(1CC)-C(1CC)-O(2CC).

Authors should explain that the CO2 occupancies have been fixed globally per molecule – do all the atoms refine to similar occupancies if refined freely and independently?

Response: According to the reviewer's suggestion, we added the refinement detail about the CO₂ occupancies of Zn-DPA•2CO₂ crystal in the supporting information. As the sorption measurements showed this compound adsorbed 2CO₂ molecules per Zn_{1.5} under 101 kPa CO₂ atmosphere at the measurement temperature, we can expect the sum total number of CO₂ at A, B, C, and CC (site I, III, II) to be two. Therefore, free variables were introduced for the occupancy refinements of all CO₂ molecules with using SUMP instruction. When SUMP instruction was not used and refined freely, the similar results were obtained with the slight overestimation of the CO₂ site occupancies (around 2.4 CO₂ per Zn_{1.5}).

The same esd has been used on DFIX 1,2 and 1,3 restraints (0.001). the 1,3 bonds restraints ESDs should be larger than those of 1,2 bonds.

Response: The ESD has been corrected as “0.02” for DFIX, “0.04” for DANG.

TEMP instruction missing.

Response: TEMP instruction has been used in the new crystal data.

Efficacy of outgassing procedure is not reported. The crystal loaded with CO₂ does not seem to be the same crystal that was used to determine the ‘outgas’ empty structure. Can you be sure that under these conditions (shorter outgassing time) the crystal is fully desolvated. It can be much harder to outgas mounted single crystals under vacuum due to poor thermal conduction.

Response: According to the reviewer’s suggestion, we added the experimental details in the supporting information. As described above, the detailed information for outgassing procedure is as follows: one single crystal was put into a glass capillary and evacuated at 120 °C under reduced pressure (below 10⁻² Pa) overnight. The glass capillary was sealed using small torch flame with keeping vacuum condition. Accordingly, we have added the related information in `_diffn_crystal_treatment` of the new CIF. And the obtained Zn-DPA-empty crystal data under the above condition displayed that there is no large Q residues (≤ 0.72) in the cavity. Meanwhile, their TGA also clearly indicated that the guest water molecules were completely excluded (Supplementary Fig. S8).

The refinement of CO₂-I is good – there is compelling evidence it is there and the refinement is appropriate and plausible.

Sites II and III are problematic. The ISOR restraint applied to their ADP makes them to all purposes isotropic – refining them as isotropic makes no difference to the R1 value of the structure. Sites II and III have much larger Ueq values than site I. This calls into question their refined occupancies of 0.21 and 0.25. If they are refined with isotropic parameters fixed as plausible values of 0.15 then their occupancies drop to

0.15. At this point I would dispute whether such low occupancy residues can be confidently identified.

Response: In the new crystal data, occupied CO₂ molecules have been resolved more reasonably, the refined occupancies of the CO₂ molecules were 0.77, 0.64 and 0.59 at sites I, II and III, respectively.

The reason for the use of a DAMP (1000) instruction in the final refinement job should be discussed/reported. If it is to aid convergence of the low occupancy CO₂ residues then it further calls into question whether they can be adequately modelled with this data. The structure is very poorly converging – it took 12 repeats of 25 cycles of least squares to converge. This is further evidence of a poor model for the data.

Response: DAMP (1000) instruction was not used in the new crystal data.

The manuscript describes interactions between adjacent CO₂ moieties. Firstly, their low partial occupancies mean that it is not certain that they ever occupy the same unit cell simultaneously. The very large ADP values and low occupancies of sites II and III mean that geometric interactions can only be described in very broad terms. In my opinion Site I is mutually exclusive to site II: the C...O distance of 2.33 Å is very short. The sum of the VdW radii for these atoms is 3.20 and intermolecular C...O distances in solid CO₂ are 3.11 Å. Benchmark DFT calculations for C...O T-interactions give values of around ca 2.8 Å. Such a short reported distance is not supported by this weak data.

Response: In the new crystal data, the cryptographically estimated CO₂ occupancies were improved to be 0.77, 0.64 and 0.59 at sites I, II and III, respectively. CO₂-I molecule interacts with neighbor CO₂-II and CO₂-III in T-shaped [(C(δ⁺)...O(δ⁻) = 3.90(9) Å, C1A...O2CC] and slipped parallel conformation [C...C = 3.70(3) Å, C1A...C1B; 3.26(3) Å, C1A...C1C].

REVIEWERS' COMMENTS:

Reviewer #4 (Remarks to the Author):

Crystallographic Review: Recommend for publication with minor corrections/suggestions.

I thank the authors for acting on the previous recommendations and carrying out improved single crystal diffraction experiments with more detailed reporting. The new data sets are appropriately refined and documented; they give a credible insight into the nature of CO₂ binding in this material.

The reporting of ESDs for geometric parameters in the manuscript is an improvement, however, in several instances values are incorrectly quoted with an ESD of zero e.g. p5 line 16,18; p8 line 9. In these cases, the value should be either rounded up to 1 or omitted.

The authors mention on p8 line 9 "It is significant that the CO₂ geometries are similar to that of solid CO₂". I think it is wholly expected rather than significant that bond lengths of weakly bound CO₂ molecules do not deviate significantly that of solid CO₂. Discussion of the experimentally determined CO₂ bond lengths in this work should be take into account the geometric restraints applied to them during refinement and the (understandably) large uncertainties on them.

Point-by-point response to the referees' comments

The reviewers' comments are written with *Italic font* and replies are written in **blue**.

Reply to the Reviewer #4's comments

Crystallographic Review: Recommend for publication with minor corrections/suggestions.

I thank the authors for acting on the previous recommendations and carrying out improved single crystal diffraction experiments with more detailed reporting. The new data sets are appropriately refined and documented; they give a credible insight into the nature of CO₂ binding in this material.

Response: We greatly appreciate his positive feedback and helpful suggestions.

The reporting of ESDs for geometric parameters in the manuscript is an improvement, however, in several instances values are incorrectly quoted with an ESD of zero e.g. p5 line 16, 18; p8 line 9. In these cases, the value should be either rounded up to 1 or omitted.

Response: As pointed out by the reviewer, the ESDs of zero have been omitted on page 5 line 16, 18; page 8 line 9.

The authors mention on p8 line 9 "It is significant that the CO₂ geometries are similar to that of solid CO₂". I think it is wholly expected rather than significant that bond lengths of weakly bound CO₂ molecules do not deviate significantly that of solid CO₂. Discussion of the experimentally determined CO₂ bond lengths in this work should be take into account the geometric restraints applied to them during refinement and the (understandably) large uncertainties on them.

Response: As reviewer pointed out, the bond lengths and angles of CO₂ have large uncertainties because of their disordered nature and low occupancies, we have deleted this sentence in the main text. Meanwhile, Ref. 29 was also deleted in the main text.